# Self-organizing neuromorphic nanowire networks as stochastic dynamical systems

Gianluca Milano [1] ✉, Fabio Michieletti [2], Davide Pilati [1,2], Carlo Ricciardi [2] & Enrique Miranda[3]

Neuromorphic computing aims to develop hardware platforms that emulate the effectiveness of our brain. In this context, brain-inspired self-organizing memristive networks have been demonstrated as promising physical substrates for *in materia* computing. However, understanding the connection between network dynamics and information processing capabilities in these systems still represents a challenge. In this work, we show that neuromorphic nanowire network behavior can be modeled as an Ornstein-Uhlenbeck process which holistically combines stimuli-dependent deterministic trajectories and stochastic effects. This unified modeling framework, able to describe main features of network dynamics including noise and jumps, enables the investigation and quantification of the roles played by deterministic and stochastic dynamics on computing capabilities of the system in the context of physical reservoir computing. These results pave the way for the development of physical computing paradigms exploiting deterministic and stochastic dynamics in the same hardware platform in a similar way to what our brain does.

In the era of Artificial Intelligence (AI) and Big Data, the continuous growth of computing demand is unsustainable with currently available digital processing and storage units based on the conventional von Neumann architecture[1]. In the race towards future technologies, neuromorphic computing aims to take inspiration from the effectiveness and advanced functionalities our brain offers to develop energy-efficient hardware platforms[2–4]. This requires the development of radically new physical substrates as well as novel data storage and communication protocols that leverage new physical phenomena for computing in the analog domain at the matter level[5] while embracing stochasticity, in a similar fashion our brain does[6]. With the aim of emulating the principle of self-organization typical of biological neuronal systems, self-organizing neuromorphic nanoscale networks have been recently demonstrated as feasible substrates for physical processing of information directly at the matter level[7–20]. Information processing and computing capabilities of these complex systems are inherently related to network dynamics, where the internal state of the system evolves over time through an adaptive behavior relying on the interaction of nano-elements driven by time-dependent external signals coming from the environment[21–23]. In these self-organizing systems, the concept of emergent behavior has been related to the collective response of a large number of nano-objects subjected to mutual interactions[21,24,25]. In opposition to classical algorithmic computation, where rules are explicitly given by a computer program, in these dynamical systems information processing relies on the underlying physical laws governing the connectivity of the nano elements[26,27]. In this context, voltage-driven deterministic dynamics occurring in memristive complex networks based on nanowires (NWs) have been exploited to emulate fundamental features of biological systems, including short-term plasticity, heterosynaptic plasticity, working memory, metaplasticity and memory engrams[12,28,29]. The associated deterministic dynamics have been exploited so far to solve a wide range of computational tasks including pattern recognition and time-series prediction in the framework of reservoir computing[30–32].

[1]Advanced Materials Metrology and Life Sciences Division, INRiM (Istituto Nazionale di Ricerca Metrologica), Torino, Italy. [2]Department of Applied Science and Technology, Politecnico di Torino, C.so Duca degli Abruzzi 24, Torino, Italy. [3]Departament d'Enginyeria Electrònica, Universitat Autònoma de Barcelona (UAB), Cerdanyola del Vallès, Spain. ✉e-mail: g.milano@inrim.it

Moreover, stochastic spiking dynamics of self-assembled percolating networks have been considered for true random number generation[33,34]. However, beyond these remarkable achievements in the field, a unified mathematical framework describing both deterministic and stochastic behaviors of self-organizing neuromorphic nanoscale networks is currently missing.

In this work, we report on the modeling of nanowire networks as dynamic systems endowing deterministic and stochastic behaviors. We show that our modeling approach describing network dynamics as an Ornstein-Uhlenbeck (OU) process with random perturbations can describe the experimentally observed evolution of the system according to an external control variable, in our case the applied voltage. In particular, the proposed compact model can describe both deterministic conductance transients induced by modifications of the applied voltages as well as stochastic conductance fluctuations including noise and jumps. The model is exploited to investigate the impact of deterministic and stochastic dynamics on the information processing capabilities of the system by considering benchmark nonlinear autoregressive moving average (NARMA) and nonlinear transformation (NLT) computing tasks. The proposed description represents a step forward in the development of neuromorphic

systems that, besides deterministic dynamics, endow stochasticity in a similar fashion to biological systems.

## Results

### Memristive network behavior

Self-assembled Ag NW networks (Fig. 1a, details of fabrication in "Methods") are complex dynamical systems, where the propagation of an electrical signal through the network is determined by Kirchhoff's laws and memristive nonlinear interactions among a huge number of NWs at nanoscale crosspoint junctions (Fig. 1b). Here, the physical mechanism of memristive activity relies on the electric-field driven dissolution and migration of $Ag^+$ ions across the insulating polymeric shell layer that surrounds the Ag NW cores. This forms a localized Ag conductive bridge at the crosspoint junction, as schematized in Fig. 1c[12]. In our case, the switching mechanism is volatile-type since Ag conductive filaments can spontaneously break down after formation with a characteristic lifetime that depends on the experienced electrical excitation, as discussed in previous works[35–37]. Besides memristive behavior of NW junctions, it is worth mentioning that resistive switching in the Ag NW itself has been experimentally observed[38]. The interaction among a large number of memristive structures, where the conductance is regulated by the interplay between filament formation

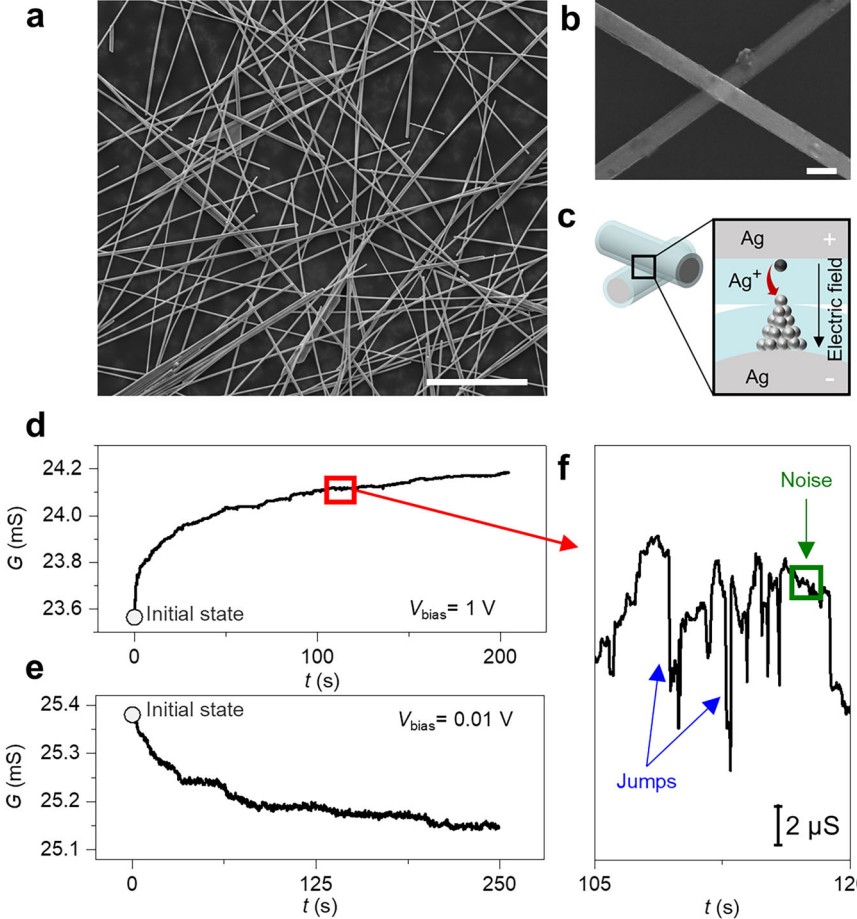

**Fig. 1 | Self-organizing neuromorphic NW networks. a** Scanning electron microscopy (SEM) image of a self-organizing neuromorphic network based on highly interconnected NWs (scale bar, 5 μm) and **b** detail of a NW junction (scale bar, 200 nm). **c** Schematic representation of the resistive switching mechanism at NW junctions, where the formation/rupture of a metallic Ag conductive filament connecting the metallic NW cores under the action of the applied electric field modulates the junction conductivity. **d** Potentiation of the neuromorphic network conductance $G$ over time, characterized by the progressive enhancement of the

overall NW network conductance from an initial state towards a higher conductance state under two-terminal constant voltage stimulation of 1 V. **e** Relaxation of the neuromorphic network characterized by a progressive decrease of the overall NW network conductance from an initial state towards a lower conductance state under two-terminal constant voltage stimulation of 0.01 V. **f** Detail of the conductance trace reported in (**d**), showing that, besides deterministic potentiation/relaxation, the neuromorphic behavior endows stochastic effects characterized by conductance fluctuations (noise) and jumps.

and spontaneous rupture, gives rise to a deterministic behavior of the system that was exploited in the past for neuromorphic-type data processing and unconventional computing[30]. The deterministic behavior is characterized by a nonlinear dynamics that, depending on the electrical stimulation and the system's initial condition, can lead to a progressive enhancement (potentiation) or decrement (relaxation) of the overall effective conductance value in between two network areas. Network conductance dynamically evolves over time due to multiple series and parallel current pathways that are sequentially formed and destroyed depending on the input stimulation and the strength of local connections[23]. In this context, experimental conductance time traces acquired under electrical stimulation results from collective phenomena and interactions that are hidden to the external observer[19]. Fig. 1d reports an example of network potentiation under constant voltage stimulation of 1 V, showing a progressive increase of conductance $G$. On the other hand, Fig. 1e illustrates an example of network relaxation under constant voltage stimulation of 0.01 V, where the conductance of the network progressively decreases towards a lower value. These dynamics rely on the evolution of the system from an initial conductance state towards a new equilibrium state determined by the applied bias voltage. Besides the observed deterministic potentiation/relaxation behavior, the investigated electrical network endows a random component characterized by low-level conductance fluctuations and jumps that cannot be overlooked (see Fig. 1f). This is the result of randomly distributed switching events at the memristive NW junctions caused by the local rearrangements of potential drops and the inherent stochastic nature of the conductive filament formation and rupture processes[39,40]. While noise effects can be attributed to conductance fluctuations in junctions distributed across the network, jumps in the conductance trace seem to correspond to transitions caused by resistive switching events in one (or few) junctions located in highly relevant topological areas of the network connecting/disconnecting entire network domains[21]. In this scenario, jumps are expected to have the same physical origin as the low-level fluctuations but can be considered rare events in terms of occurrence probability and magnitude.

### Memristive networks as stochastic dynamical systems

A unified framework based on an OU process with jumps is here exploited for modeling deterministic and stochastic dynamics of neuromorphic nanowire networks, as described in the following. The OU process with jumps[41] is described by an Itô-type differential equation, which involves the combined action of deterministic and stochastic terms. The variable considered in this approach is the internal memory state of the system $g$ (i.e., normalized conductance, $0 \le g \le 1$) that evolves over time depending on the history of applied electrical stimulation. The evolution of $g$ can be described by the stochastic differential equation (SDE):

$$\frac{dg}{dt} = \underbrace{\theta[\widetilde{g} - g]}_{deterministic} + \underbrace{\sigma dW}_{noise} + \underbrace{\Gamma dq}_{jumps} \tag{1}$$

where $\widetilde{g}$ represents the long-term mean or equilibrium memory state (steady state), $\theta$ the reversion speed (i.e., the rate at which $g$ reverts towards $\widetilde{g}$), $\sigma$ the noise intensity (assumed independent of $g$), $dW$ the Gaussian noise (Wiener process), $\Gamma$ the jump amplitude, and $dq$ the jump occurrence rate. It is worth emphasizing that assuming a constant $\sigma$ does not mean that stochastic and deterministic dynamics are independent, since the solution of Eq. (1) relies on the noise and jumps effects in combination with the (deterministic) mean-reverting process.

Originally developed as a model for describing the velocity of a Brownian particle under the influence of friction (Langevin equation[42]), the OU process which is contemporarily a Gaussian and a Markov

process, has been exploited to model stochastic dynamical systems in a wide range of contexts such as financial systems and natural sciences[43]. This represents the simplest Markov-Gaussian process that can be postulated, where coupling of deterministic and stochastic components is inherent to the assumed dynamics. In a first-order approximation, the current $I$ that flows through the network (i.e., the physical observable) when a voltage $V$ is applied across two network areas relates to the internal memory state of the system through Ohm's law as:

$$I = \left[ G_{\min}(1 - g) + G_{\max} g \right] \cdot V = G \cdot V \tag{2}$$

$G_{\min}$ and $G_{\max}$ are the minimum and maximum conductance values while $G$ is the network conductance which depends on $g$.

### Deterministic behavior of the neuromorphic network

The deterministic behavior of the memristive network has been experimentally analyzed by collecting the time traces of the conductance $G$ when stimulated with a fixed voltage bias. Figure 2a reports experimental traces of the time-dependent evolution of $G$ from the initial network ground state (i.e., stable state when not stimulated) towards a new steady state when biased with different voltages ranging from 0.1 V up to 6.6 V (experimental details in "Methods"). The progressive increase of conductance over time while applying a constant bias voltage is related to the self-organized formation and subsequent consolidation of conductive pathways formed by activated junctions that bridge stimulated network areas, as previously investigated through both experiments[29] and modeling[23]. The acquired experimental dataset allows us to experimentally investigate the dependence of the steady state $\widetilde{G}$ as a function of the applied voltage, as reported in Fig. 2b. $\widetilde{G}$ is The value corresponding to the long-term stabilization of the network state (details in the inset). Experimental results show a sigmoidal-like transition of the steady state conductance from a low to a high $\widetilde{G}$ value as a function of the applied voltage. This deterministic trajectory of the network can be modeled using a potentiation-depression rate balance equation[44] that represents the deterministic form of the memory state Eq. (1) ($\sigma = 0$ and $\Gamma = 0$) expressed as:

$$\frac{dg}{dt} = \underbrace{\theta[\widetilde{g} - g]}_{deterministic} = \kappa_{\mathrm{P}}(1 - g) - \kappa_{\mathrm{D}} g \tag{3}$$

$\kappa_{\mathrm{P}}$ and $\kappa_{\mathrm{D}}$ are potentiation and depression rate coefficients which exponentially depend on the applied voltage through physics-based relationships accounting for the forward/backward diffusive ionic processes occurring at the NW junctions:

$$\kappa_{\mathrm{P}}(V) = \kappa_{\mathrm{P0}} \exp\left( + \eta_{\mathrm{P}} V \right), \kappa_{\mathrm{D}}(V) = \kappa_{\mathrm{D0}} \exp\left( - \eta_{\mathrm{D}} V \right) \tag{4}$$

where $\kappa_{\mathrm{P0}}, \kappa_{\mathrm{D0}} > 0$ are constants and $\eta_{\mathrm{P}}, \eta_{\mathrm{D}} > 0$ transition rates. Since a single rate-balance equation is used to describe the dynamic behavior of the entire network, transition rates here represent effective network parameters. Notice, from Eq. (3), that the reversion speed $\theta$ and the equilibrium state of the system $\widetilde{g}$ are both functions of the applied voltage $V$ according to the expressions:

$$\theta(V) = k_{\mathrm{p}} + k_{\mathrm{d}} \tag{5}$$

$$\widetilde{g}(V) = \frac{k_{\mathrm{p}}}{k_{\mathrm{p}} + k_{\mathrm{d}}} \tag{6}$$

In Eq. (6), $\widetilde{g}$ the internal memory of the network in steady-state conditions is represented. Importantly, Eq. (3) cannot only be solved numerically through the Euler method, but also analytically, following

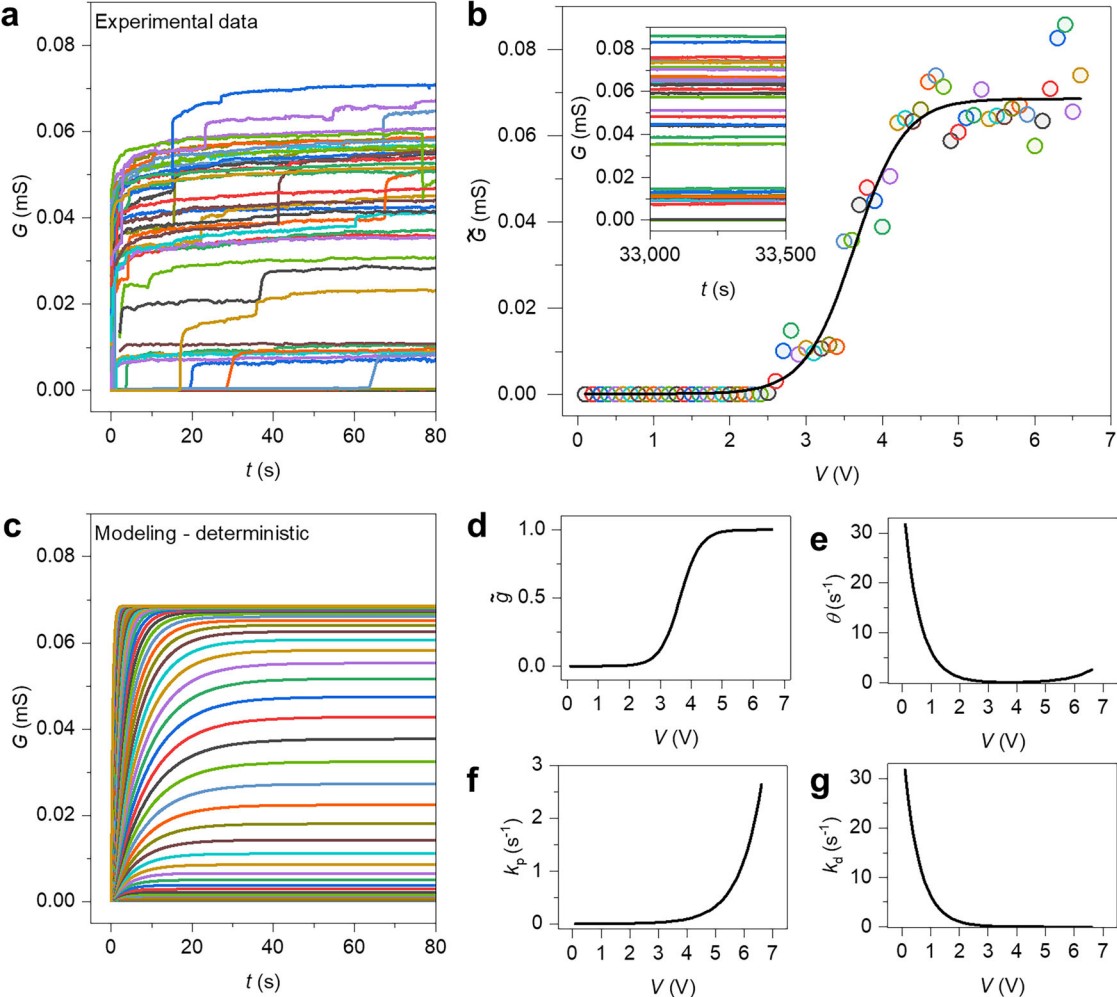

**Fig. 2 | Experimental and modeling deterministic dynamics of neuromorphic NW networks. a** Experimental time traces of the conductance $G$ under constant voltage stimulation, where each curve represents the evolution of $G$ over time from the initial ground state towards the new equilibrium state for different applied voltages (from 0.1 V to 6.6 V, step of 0.1 V). **b** Experimental (circles) and modeled (line) evolution of the stationary steady state ($\widetilde{G}$) as a function of the applied voltage bias. The inset shows the time trace of the conductance after 33,000 s of constant voltage bias application, showing that the conductance state is stationary. Circle colors correspond to the line colors of experimental time traces reported in the inset and in (**a**). **c** Modeling of deterministic network dynamics for different applied voltages (from 0.1 V to 6.6 V, step of 0.1 V). Line colors correspond to the color of the experimental time traces obtained with the same applied voltage reported in (**a**). **d** Internal memory steady state $\widetilde{g}$, **e** reversion speed $\theta$, **f** potentiation rate coefficient, and (**g**) depression rate coefficient as a function of the applied voltage derived from modeling.

a recursive approach (details in "Methods"). The experimental dependence of the steady state reported in Fig. 2b can be well interpolated by the potentiation-depression rate-balance model through Eq. (6). The interpolation of experimental steady states that enables the retrieval of rate coefficients and transition rates of the network allows also inferring the deterministic network transient dynamics, as reported in Fig. 2c. By comparing modeling with experimental results reported in Fig. 2a, it can be observed that the proposed model describes quite well the main features of the conductance evolution over time during transients (i.e., before reaching the steady state condition). The memory steady state $\widetilde{g}$, the reversion speed $\theta$, the evolution of the potentiation and depression rate coefficients $\kappa_{\mathrm{p}}$, and $\kappa_{\mathrm{d}}$, as a function of the applied voltage derived from interpolation of the experimental data reported in Fig. 2b, are illustrated in Fig. 2d, e, f, g, respectively. It is worth mentioning that in the context of system dynamics, $\widetilde{g}$ it represents the stable trajectory of the system, i.e., the statistically invariant phase the system reaches for a given voltage, irrespective of the initial conducting state (mean-reverting property of the OU process). Even if experimental observations suggest that the system's conductance tends toward the same steady state irrespective

of the initial condition (details in Supplementary Fig. 1), a wider variety of starting conditions would be required to demonstrate unequivocally the occurrence of hard attractor states. Notably, the proposed compact description endowing mean-reverting property enables us to describe the experimentally observed dependence of the network state only on the recent history of applied stimulation, a property that has been exploited for temporal-processing of the input signal in the framework of reservoir computing[30].

## Stochastic behavior of the neuromorphic network

The stochastic effects in the neuromorphic networks were analyzed by disentangling noise and jumps in the conductance time trace when the system operates in the stationary state[45]. It is worth emphasizing once again that, according to the chosen stochastic process, deterministic behavior, noise, and jumps are part of the whole system's trajectory even under steady state conditions. Figure 3a reports experimental changes in the conductance $dG/dt$) monitored for more than 15,000 s with an applied voltage of 3.6 V. As can be seen, small conductance fluctuations related to noise and large spike events related to conductance jumps occur. Consequently, the corresponding $dG/dt$

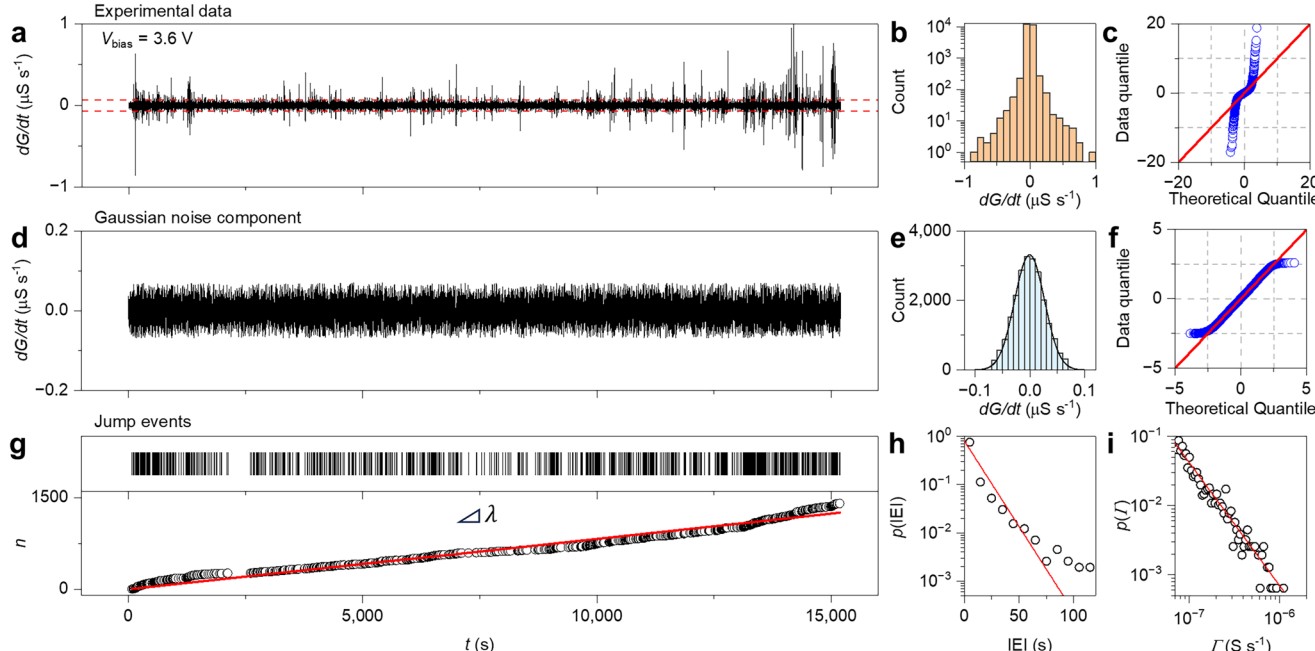

**Fig. 3 | Experimental stochastic dynamics of neuromorphic NW networks stationary state. a** Experimental changes in the conductance $dG/dt$ over time of the NW network in the stationary state sustained by an applied constant voltage of 3.6 V (red dashed line represents the calculated threshold value for noise disentanglement), **b** histogram of $dG/dt$ and **c** corresponding quantile-quantile plot of experimental data against the theoretical quantiles of a normal distribution revealing the presence of heavy tails. **d** Experimental Gaussian noise component of the signal obtained by noise disentanglement, **e** histogram of the Gaussian component of $dG/dt$ fitted with a Gaussian distribution (black line), and

**f** corresponding quantile-quantile plot of experimental data against the theoretical quantiles of a normal distribution. **g** Experimental raster plot of jump events (top panel) and number of jump events $n$ as a function of time (low panel), where experimental data (circles) are interpolated by a straight line with slope $\lambda$ (event rate or intensity). **h** Experimental probability distribution of inter-event intervals (IEIs) (circles) and the theoretical exponential distribution of IEIs expected in case of a homogeneous Poisson process with intensity $\lambda$ (red line). **i** Experimental probability distribution for the jump amplitude $\Gamma$ that can be interpolated by a power law distribution (red line).

histogram reported in Fig. 3b shows a heavy tailed distribution, as also revealed by the quantile-quantile plot assuming a normal distribution (Fig. 3c). The obtained results indicate that a large proportion of the detrended data is located at the tails of the distribution in comparison with what is expected for the normal case. The disentanglement of noise and jump events was in practice performed through a thresholding algorithm that maximizes the p-value of the Gaussian distribution of the noisy component of the signal (details in "Methods," the calculated threshold value is reported in Fig. 3a). Figure 3d reports the experimental noise component of the stationary state signal over time disentangled from jumps, where the normal distribution of the signal (Fig. 3e) and the quantile-quantile plot (Fig. 3f) reveals the Gaussian nature of the fluctuating component (presumably because of the central limit theorem). Remarkably, Gaussian behavior is directly related to the network activity and is not linked to the experimental setup and/or measurement protocol (details in Supplementary Fig. 2). Figure 3g reports the experimental raster plot of jump events and their corresponding evolution over time $n(t)$, showing that $n(t)$ increases almost linearly with slope $\lambda \sim 0.082$ events per second. In case of a homogeneous stochastic Poisson process, $\lambda$ represents the event occurrence rate or intensity of the process. In this case, the interarrival times between events (interevent intervals, IEIs) are independent and identically distributed, where the density distribution of IEI can be described through the exponential distribution $p(\text{IEI}) = \lambda e^{-\lambda \text{IEI}}$. As can be observed in Fig. 3h, the experimental density distribution of IEIs is in good agreement with the theoretical distribution expected for a Poisson process. Together with the linear increase of the number of events over time, these results suggest that the occurrence of jumps can be well described by a homogeneous Poisson process, i.e., a stochastic process where: (i) the average rate (events per period) is constant, (ii) events are independent of each other, and (iii) two events

cannot occur at the same time. It is worth mentioning that the Poisson process exploited for modeling jumps does not provide temporal correlation between jump events, as for example expected in networks operating in the critical state[20,46]. Note that a Poisson process has also been used to describe the probability of switching events in conventional memristive cells[39]. Furthermore, the experimental probability distribution of jump amplitudes reported in Fig. 3i follows a power law distribution $p(\Gamma) \propto \Gamma^{\alpha}$, where $p(\Gamma)$ denotes the probability of an event with amplitude $\Gamma$. The exponent obtained from experimental data is $\alpha \sim -2.78 \pm 0.07$ (details in "Methods").

The stochastic behavior of the network in the stationary state ($g = \tilde{g}$) can be described through the stochastic form of the memory state Eq. (1). While this equation has analytic solution (stochastic) in case of no jumps ($\Gamma = 0$), when jumps are included, the stochastic differential equation needs to be solved numerically through the Euler–Maruyama method (details in "Methods"). Note that modeling noise with a Wiener process is consistent with experimental results showing Gaussian dispersion. Therefore, based on experimental results and previous discussions, stochastic jumps can be modeled through a homogeneous Poisson point process. In this case, spike generation fulfils the relationship $n(t) = \lambda t$, where $\lambda$ represents the event rate of the Poisson process ($\lambda \sim 0.082$ events per second according to experimental results). In this context, it is possible to generate Poisson spike events on the fly, where the probability of observing a jump is given (for small $\delta t$) by $p\{1 \text{ spike during } \delta t\} \approx \lambda \delta t$[43]. According to experimental results, the amplitude $\Gamma$ of each jump follows a power-law distribution with exponent $\alpha = -2.78$ (details in "Methods"). Results of modeling the stochastic behavior of the NW network experimentally reported in Fig. 3 are shown in Fig. 4 (details of model calibration in "Methods"). As can be observed by comparing Figs. 3 and 4, the model correctly

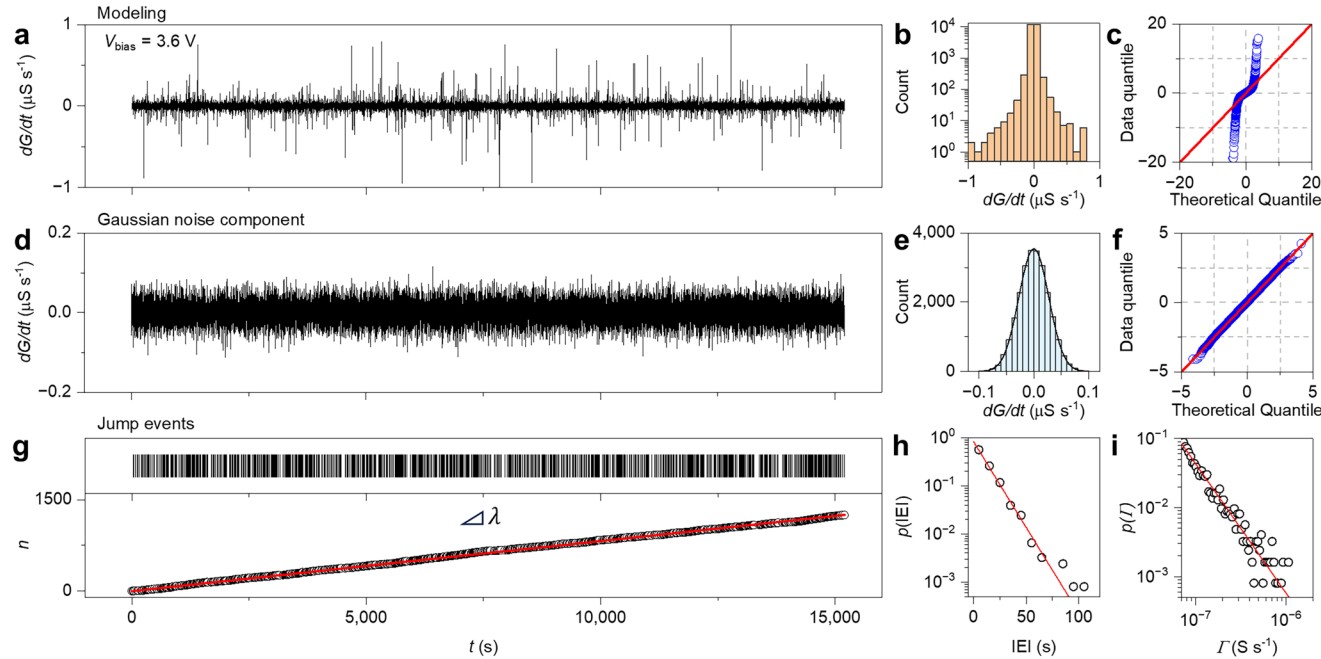

**Fig. 4 | Modeling stochastic dynamics of neuromorphic NW networks stationary state. a** Modeled changes in the conductance $dG/dt$ over time of the NW network in the stationary state sustained by an applied constant voltage of 3.6 V, **b** histogram of $dG/dt$ and **c** corresponding quantile-quantile plot of modeled data against the theoretical quantiles of a normal distribution, revealing the presence of heavy tails similar to experimental data. **d** Experimental Gaussian noise component of the modeled signal, **e** histogram of the Gaussian component of $dG/dt$ fitted with a Gaussian distribution (black line), and **f** corresponding quantile-

quantile plot of modeled data against the theoretical quantiles of a normal distribution. **g** Experimental raster plot of modeled jump events (top panel) and number of jump events $n$ as a function of time (low panel), where modeled data (circles) are interpolated by a straight line with slope $\varepsilon$. **h** Probability distribution of inter-event intervals (IEIs) obtained from modeling (circles) and the theoretical exponential distribution of IEIs expected in case of a homogeneous Poisson process with event rate $\lambda$ (red line). **i** Simulated probability distribution of jump amplitude $\Gamma$ that can be interpolated by a power law distribution (red line).

addresses the experimental changes of the conductance ($dG/dt$) both in terms of time trace (Fig. 4a, additional data in Supplementary Fig. 3) and distribution (Fig. 4b, c). The intertwined action of stochastic and deterministic effects endowed in our modeling approach results also in qualitative agreement with the experimental and modeled conductance time traces in the stationary state (Supplementary Fig. 4). Despite further experiments are required to elucidate the coupling between deterministic and stochastic dynamics in the physical system, the OU modeling approach is in agreement with (i) the experimental observation of the reversion to the average trajectory after stochastic jumps in the experimental conductance time trace (examples in Supplementary Fig. 4) and (ii) the exponential decay of the autocorrelation function with the number of lags in the stationary state as expected for an OU process (Supplementary Fig. 5).

Even if the OU modeling approach can represent the simplest approximation of the actual behavior of the experimental system, the model statistically well describes stochastic effects, including the time-trace and distribution of the Gaussian noise component (Fig. 4d–f, respectively), as well as the total number of jumps at time $t$, $n(t)$ (details in Supplementary Fig. 6), the probability distribution of IEIs, and the jump amplitude distribution for $\Gamma$ (Fig. 4g–i, respectively).

## Deterministic and stochastic dynamics

As discussed in previous sub-sections, resultant dynamics of NW networks can be described through the stochastic differential Eq. (1), which is able to encompass the action of deterministic and random behaviors. As an example, Fig. 5a reports the experimental trajectory of a NW network including transient effects, Fig. 5b the deterministic modeling, and Fig. 5c the stochastic modeling, including, besides the deterministic behavior, noise and jump events (details in "Methods"). While deterministic modeling with mean-reverting property can capture the average features of the conductance dynamics, including

transients, the stochastic component allows addressing the deviations that arise from the local activity of the NW network's junctions.

## Potential landscape of network dynamics

Since the neuromorphic NW network can be modeled as a stochastic dynamical system characterized by voltage-dependent trajectories, it is worth exploring its behavior in terms of the potential landscape function $U$ obtained from the deterministic form of Eq. (1), where $dg/dt = -\partial U/\partial g$ (details in "Methods"). The potential landscape of the neuromorphic network as a function of the normalized internal state of the network $g$ and applied voltage $V$ is illustrated in Fig. 6a (here, the white dashed line represents the minimum of $U$ as a function of $g$ and $V$). In this context, stable states can be conceptualized as basins in the potential landscape, where the basin depth indicates the state stability (i.e., the amount of external force needed to alter the internal memory state of the system). Importantly, Fig. 6b shows that the potential landscape of the dynamical system changes according to the applied voltage. Since we are dealing with a linear dynamical system, the potential profile exhibits parabolic shape at fixed bias with its minimum (stable state) shifting progressively from 0 to 1 as the voltage is increased (phase portrait of the system in Supplementary Fig. 7). This shift in the stable state is consistent with the sigmoidal-like transition of the experimental stationary conductance state ($\widetilde{G}$) reported in Fig. 2b (and corresponding evolution of the normalized internal memory state reported in Fig. 2d). This means that, in the proposed description, neuromorphic dynamics of self-assembled networks arising from variations in the applied voltage results in a change of the potential profile of the dynamical system over time, driving the system towards a new stable state. Due to the parabolic shape of the potential landscape, no bifurcations or transitions among coexisting deterministic stable states are expected to occur in the proposed modeling approach. Note that more complex potential landscapes caused by the

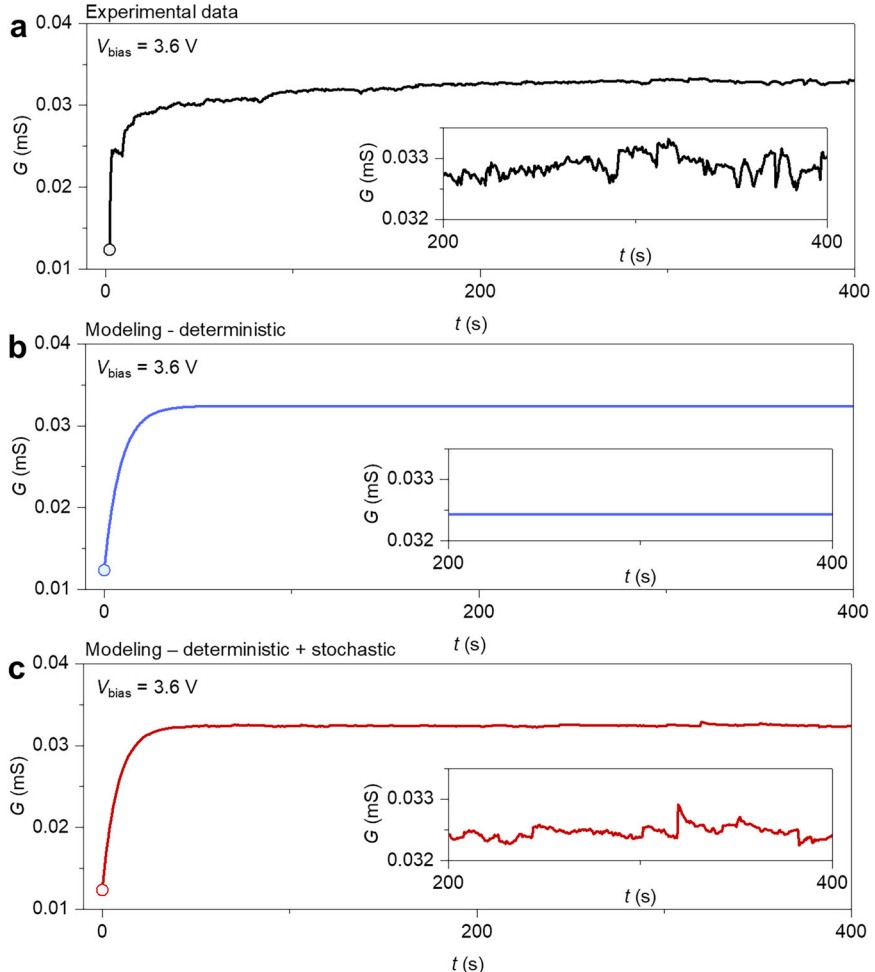

**Fig. 5 | Modeling stochastic dynamics of neuromorphic NW networks.**
**a** Experimental dynamics of the conductance of a NW network initially in the ground state (circles) under constant voltage stimulation of 3.6 V, **b** dynamics by deterministic modeling, and **c** dynamics by stochastic modeling. Insets show a detail of conductance fluctuations.

application of strategically located external biases in a multiterminal configuration would yield alternative dynamics. In our case, transient dynamics among stable states are due to changes in the applied voltage, resulting in neuromorphic potentiation/relaxation behavior of the system. By considering a stationary state, noise represents fluctuations of the stochastic dynamical system around the potential basin, while jumps correspond to sudden deviations of the internal state of the system over time from $U(t)$ to $U(t + \delta t)$. This is illustrated in Fig. 6c, where a stationary state sustained by 3.6 V is considered (enlarged view of the time trace in Supplementary Fig. 8). Here, it is possible to observe through modeling that noise and jumps allow the system to displace the internal memory states around the potential basin over time (upper and intermediate panel). The occupation probability of the equilibrium memory state is represented by the histogram shown in the bottom panel, which at the end is the stationary solution of the Fokker-Planck equation[43].

## Relationship between steady states and signal processing

Information processing in neuromorphic nanoscale systems occurs by encoding inputs from the environment into their internal state and processing through state dynamics. In this context, the external stimulus is usually transformed into time-dependent voltage input signals to be applied to the network, while information processing occurs by exploiting the conductance transients induced by internal voltage rearrangements and short-term memory effects. In our proposed

modeling approach, this means that information processing capabilities arise from the fluctuation of the system around a steady state condition induced by the electrical input signal. However, also the internal dynamics of the modeled network relies on the applied bias voltage since different voltages lead to different reversion speeds $\theta$ of the OU process (Supplementary Fig. 9). Note that a high reversion speed means that the output signal immediately responds to the input signal, while a low reversion speed is typical of a structure with a high inertia, i.e., a high resistance to changes. In this framework, the selection of an appropriate operational regime is crucial for optimizing the network response to a given input signal.

Figure 7 illustrates an example of how our model allows us to analyze the effect of deterministic and stochastic dynamics on the evolution of the internal memory state of the network. Figures 7a and 7b report the modeled deterministic and stochastic evolution of the internal memory state of the NW network in a stationary state (i.e., following the initial transient response), when stimulated with a triangular voltage waveform (input signal), while applying different constant biases (voltage applied to sustain the network operational regime) (see "Methods", transient dynamics in Supplementary Fig. 10). In particular, Fig. 7a shows the electrical response of the network operating in a regime sustained by a constant bias of 3.6 V that drives the system near the sigmoidal-like transition of the steady state expected in stationary conditions ($\tilde{g}$ ~ 0.5). Figure 7b shows the network operating in a regime sustained by a constant bias of 5 V that

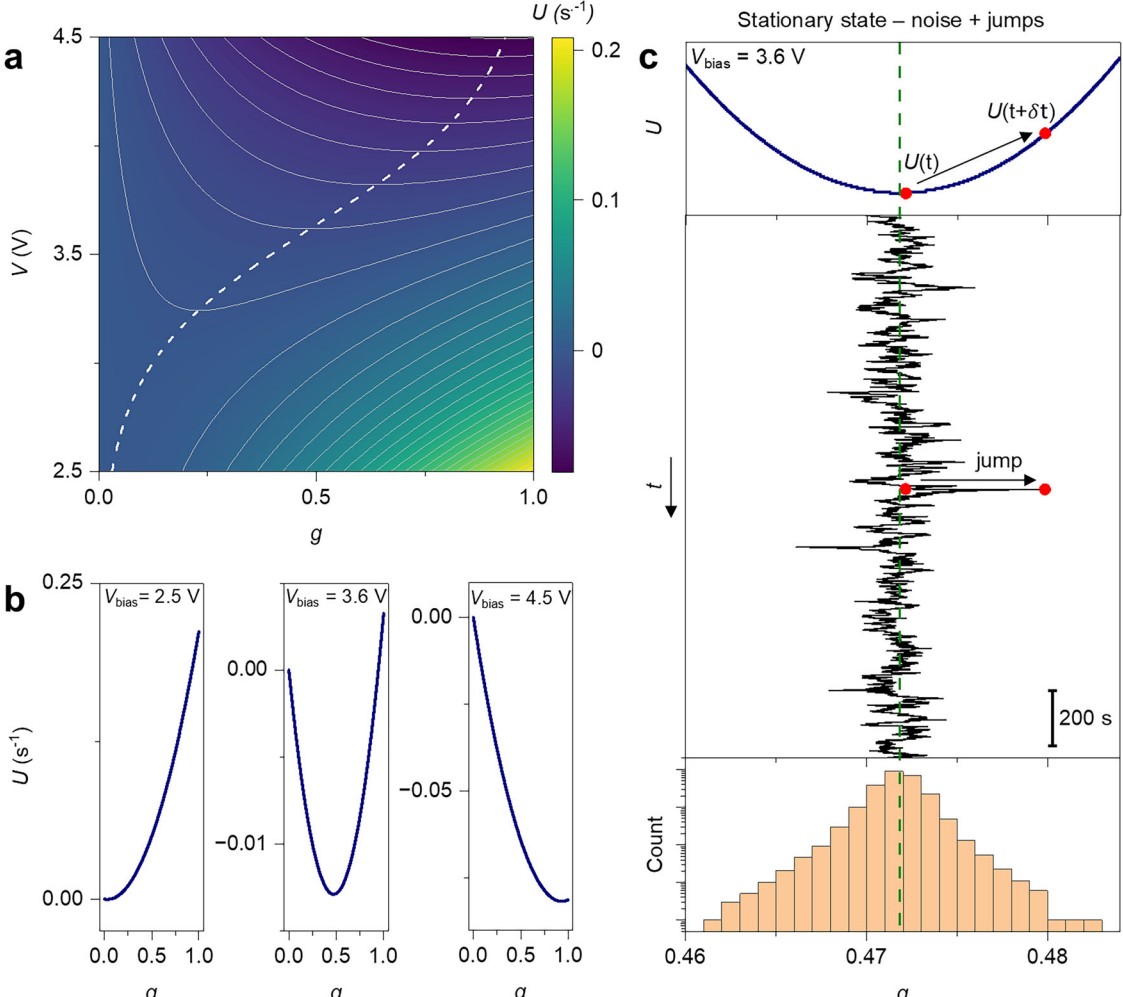

**Fig. 6 | Potential landscape of neuromorphic NW networks. a** Potential landscape as a function of the normalized internal state of the network $g$ and applied voltage $V$. The white dashed line represents the potential minimum, i.e., the asymptotic steady state of system. **b** Potential profiles at fixed voltage bias (note that $y$ scale is optimized for visualization). **c** Evolution over time of the internal memory state of the stochastic dynamical system in a stationary state sustained by an applied voltage of 3.6 V obtained by modeling. The upper panel represents the

potential profile corresponding to the stationary state, the intermediate panel represents the evolution over time of the internal memory state, and the lower panel represents the histogram of memory state occupancy obtained by monitoring the evolution of the internal state of the system for -15,000 s. While noise represents fluctuations of the system around the potential basin, jumps are represented by sudden changes in the memory state and corresponding potential over time, as represented by the arrow.

drives the system to a $\widetilde{g}$ value close to 1 (Supplementary Fig. 11). While the applied constant bias voltage is responsible for driving the system towards a steady state, the triangular voltage waveform induces fluctuations in the system state around the corresponding steady state (detailed fluctuations of the stable state in Supplementary Fig. 12). It can be observed that the network response to the same signal input is remarkably different depending on the operating regime established by the polarization bias voltage, in accordance with experimental results reported in ref. 19. Furthermore, it can be observed that enhanced dynamics of the internal memory state (i.e., higher dynamical range of $g$) is achieved when the network operates under constant bias of 3.6 V in comparison with operations at 5 V. A similar effect can be observed by considering deterministic and stochastic trajectories of the system under both operating regimes (Fig. 7c–f). Noticeably, the network dynamics is strongly affected by the stochastic effects (both noise and jumps) when operating under a 5 V constant bias (refer to Fig. 7f), while these effects are less influential when operating at 3.6 V bias. Based on our modeling approach, these results show that it is possible to tune the dynamical response of the network to an input signal by regulating the voltage-controlled stable

state representing the operating regime of the network. The magnitude of the input signal frequency in connection with the internal time response of the nanoscale system is also a factor to consider.

## Steady states and computational capabilities

Computational capabilities of self-organizing complex networks of NWs have been evaluated in the framework of the reservoir computing paradigm, where modeling can be exploited to investigate the effect of deterministic and stochastic effects on computational performance. For this purpose, a time-multiplexed reservoir computing scheme was implemented through simulations by considering a single dynamical node with delayed feedback, following the approach proposed by Appeltant et al.[47]. This implementation strategy exploits the two-terminal dynamical response of the network. It basically consists in the generation of a virtual reservoir by applying masked input signals to the dynamical system and a time-multiplexing of the network state. The mask considers $N$ virtual nodes with a separation time $\Theta = \tau/N$, $\tau$ being the delay time. In this method, a linear combination of weighted signals is passed from the reservoir to the output layer to generate the response of the system. The weights are trained using a linear

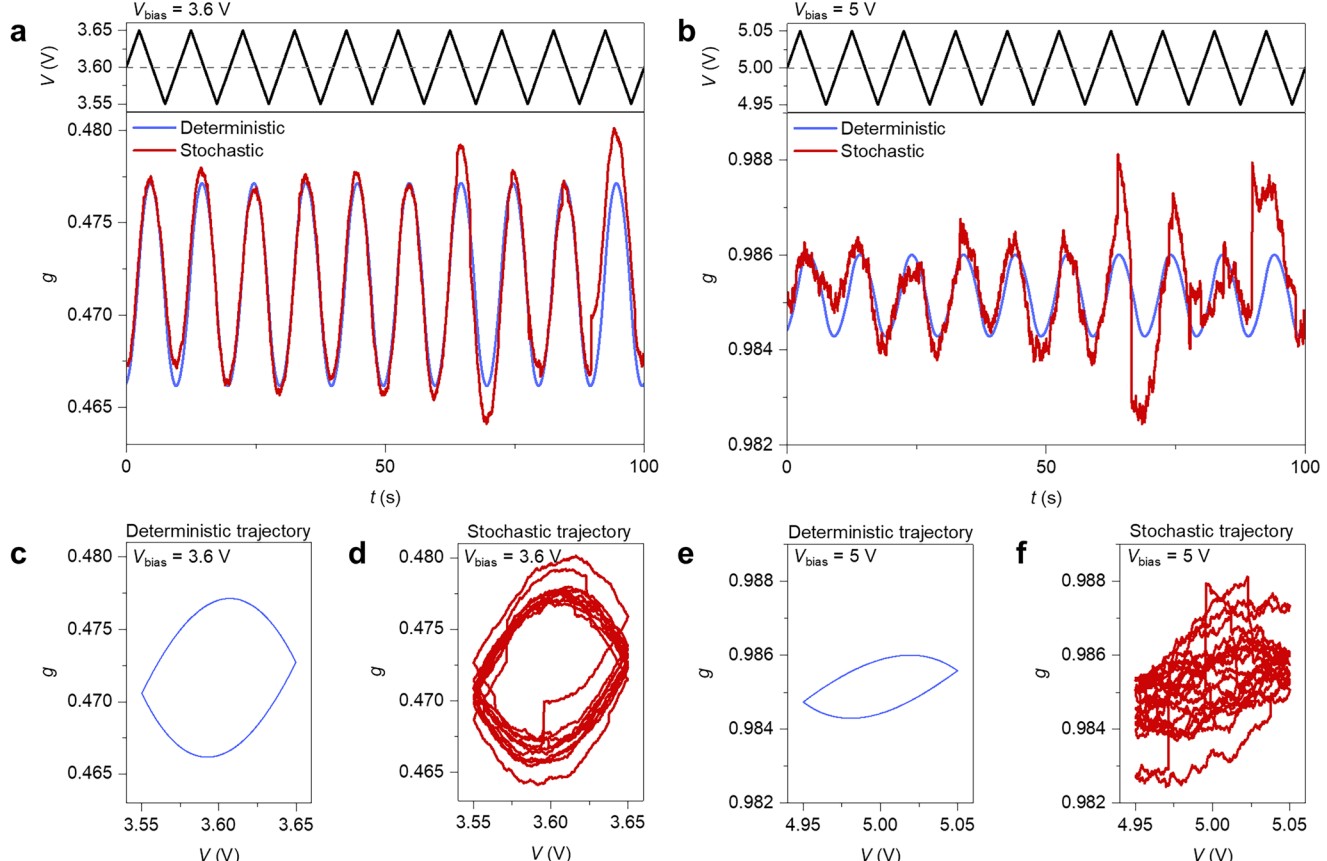

**Fig. 7 | Processing of the input signal through network dynamics.** Simulated deterministic and stochastic evolution of the internal memory state of the NW network $g$ (in stationary conditions) when stimulated with a triangular voltage waveform with amplitude of 50 mV (external signal), while applying a constant bias of **a** 3.6 V and **b** 5 V. Deterministic and stochastic trajectory of the dynamical system when operating with a voltage bias of 3.6 V (**c**, **d**) and 5 V (**e**, **f**).

regression algorithm taking into account the reservoir signals according to a target function representing the desired output (details in Supplementary Fig. 13). Computing capabilities were tested on standard benchmark tasks, namely the time series prediction corresponding to a nonlinear autoregressive moving average (NARMA) task[48] requiring both nonlinearity and memory, and Nonlinear Transformation (NLT)[49] tasks (details in "Methods"). For each considered task, computing performances of deterministic and stochastic network models were evaluated in terms of the normalized mean square error (NMSE) as a function of the bias voltage and the corresponding memory steady state $\widetilde{g}$, for different amplitudes of the input signal (details in "Methods")

Computing performances for the second-order NARMA task (NARMA-2) as a function of the applied voltage and $\widetilde{g}$ are reported in Fig. 8a, b, respectively. Here, it is shown that enhanced dynamics corresponding to $\widetilde{g} \sim 0.5$ (bias ~3.6 V) results in a minimum of NMSE, particularly evident in case of stochastic dynamics. In the case of deterministic dynamics, the increase of NMSE for $\widetilde{g} \rightarrow 0$ and $\widetilde{g} \rightarrow 1$ can be attributed to a saturation of the system response that lead to a progressive reduction of its fading memory capabilities. In the case of stochastic dynamics, this progressive reduction of fading memory is coupled with a progressive increase of the stochastic contribution that further increases the NMSE when moving away from $\widetilde{g} \sim 0.5$. While there is negligible influence of the input signal amplitude on the NMSE in the case of deterministic dynamics, when considering stochastic dynamics, a higher input signal results in a lower NMSE (Fig. 8c). This occurs because a higher input signal enhances the amplitude of the deterministic network dynamics, thus reducing the influence of the stochastic effects on the evolution of the memory state by enhancing

the signal-to-noise ratio. By considering deterministic dynamics, the prediction obtained with optimal parameters of NARMA-2 when operating the system with a bias voltage of 3.6 V ($\widetilde{g} \sim 0.5$) is reported in Fig. 8d ($N$ and $\Theta$ parameters on the NMSE is shown in Fig. 8e), while the prediction when operating the system with a bias voltage of 5 V ($\widetilde{g} \sim 0.99$) is reported in Fig. 8f ($N$ and $\Theta$ parameters on the NMSE is shown in Fig. 8e). Considering stochastic dynamics, the prediction obtained with optimal parameters of NARMA-2 when operating the system with a bias voltage of 3.6 V ($\widetilde{g} \sim 0.5$) is reported in Fig. 8h ($N$ and $\Theta$ parameters on the NMSE is shown in Fig. 8i), while the prediction when operating the system with a bias voltage of 5 V ($\widetilde{g} \sim 0.99$) is reported in Fig. 8j ($N$ and $\Theta$ parameters on the NMSE is shown in Fig. 8k).

Here, by considering deterministic dynamics exclusively, it is possible to observe that a larger window of $N$ and $\Theta$ parameters lead to low NMSE (i.e., a larger portion of dark blue in colormaps reported in Fig. 8) when operating the network at $\widetilde{g} \sim 0.5$ (Fig. 8d) with respect to $\widetilde{g} \rightarrow 1$ (Fig. 8f). When operating the network away from $\widetilde{g} \sim 0.5$, besides a general increase of NMSE, a substantial decrease in performance can be observed by considering a set of parameters with high $N$ or high $\Theta$ values, i.e., in cases where network dynamics are expected to be more affected by a reduction of fading memory properties (details in Supplementary Fig. 14).

When considering also stochastic effects, it can be observed that the parameters' range leading to low NMSE is further reduced with respect to the deterministic case (refer to Fig. 8i, k). Notably, it can be observed that performances are highly degraded for some specific choices of $N$ and $\Theta$ values. While the choice of the mask is not substantially affecting the system in the deterministic case, results show

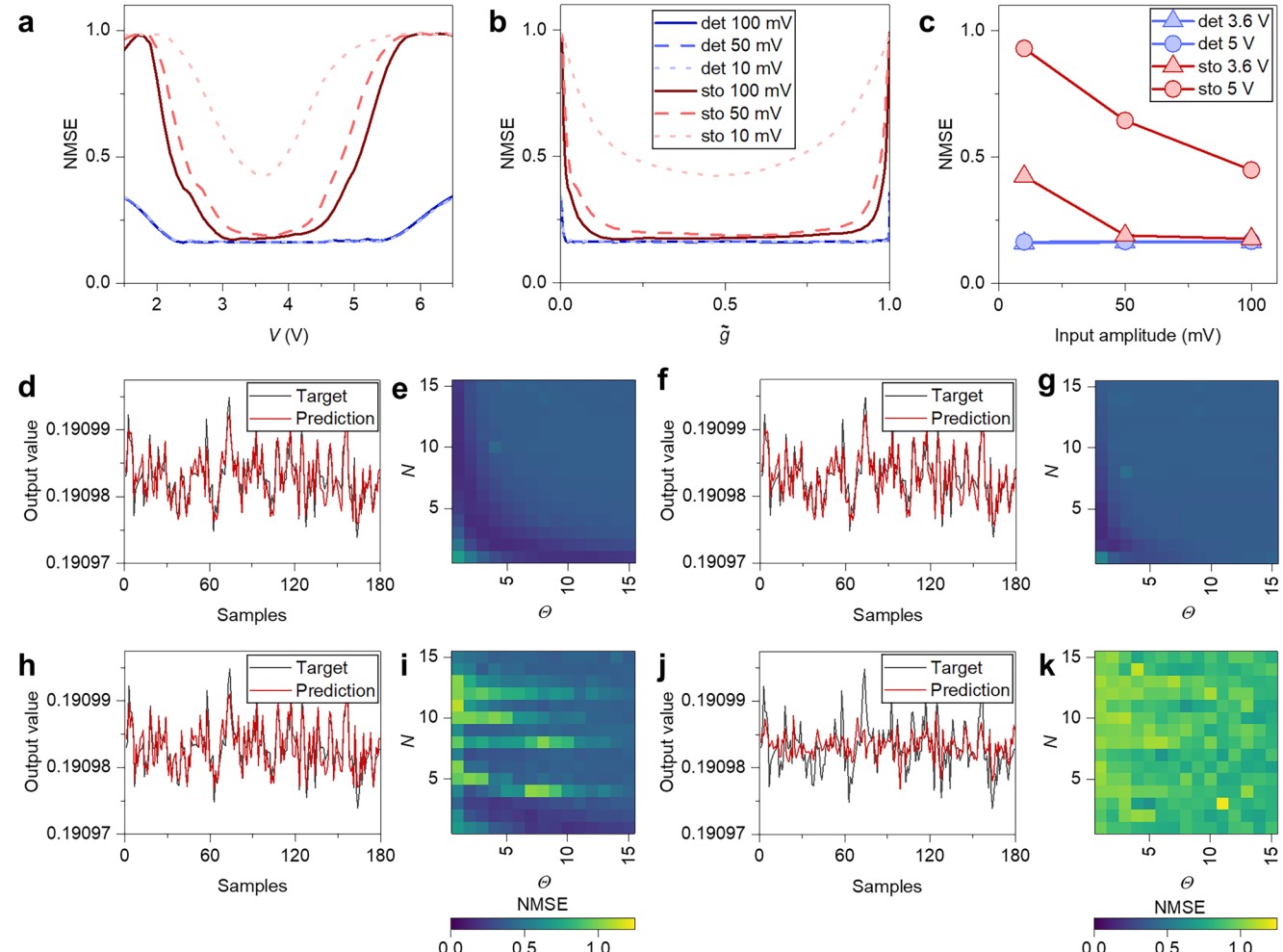

**Fig. 8 | Nonlinear autoregressive moving average (NARMA) task.** Simulation results of the NARMA-2 task in terms of NMSE as a function of **a** the bias voltage, and **b** steady state $\tilde{g}$ by considering deterministic and stochastic dynamics and for various amplitudes of the input signal. **c** NMSE as a function of the input signal amplitude for deterministic and stochastic dynamics, by considering polarization voltages of 3.6 V ($\tilde{g}$ - 0.5) and 5 V ($\tilde{g}$ - 0.99). Predictions of NARMA-2 relative to polarization voltages of 3.6 V and 5 V obtained with optimal parameters through deterministic dynamics in **d**, **f**, respectively ([$N$, $\Theta$] in **d**, **f** are [9,1] and [2,7], respectively). Colormaps showing task performances as a function of $N$ and $\Theta$ parameters for polarization voltages of 3.6 V and 5 V in **e**, **g**, respectively. Predictions of NARMA-2 relative to polarization voltages of 3.6 V and 5 V obtained with optimal parameters through stochastic dynamics in **h**, **j**, respectively ([$N$, $\Theta$] in **d**, **f** are [3,1] and [3,5], respectively). Colormaps showing task performances as a function of $N$ and $\theta$ parameters for polarization voltages of 3.6 V and 5 V in **i**, **k**, respectively. Colormaps and predictions in **d**–**k** refer to results obtained by stimulating the network with an input with an amplitude of 50 mV.

that the specific $N$ and $\Theta$ values with degraded performances rely on the specific masking scheme adopted for time multiplexing (details in Supplementary Fig. 15). This happens because each specific masking scheme combined with the input signal generates different network dynamics, driving the network to a dynamic regime that endows information processing capabilities that can be less or more resilient to stochastic effects. The worst-case scenario is obtained when considering stochastic dynamics with steady state value $\tilde{g} \rightarrow 1$ (Fig. 8k), where higher values of NMSE are observed for any $N$ and $\Theta$ value. Similar considerations apply to the results obtained from NLT tasks, as reported in Fig. 9a–f where sine to cosine wave transformations are reported (additional sine to triangular square, and sine to square wave transformations in Supplementary Fig. 16 and 17, respectively). Also in this case, the degradation of performances for specific $N$ and $\Theta$ values when considering stochastic effects relies on network dynamics generated by the specific masking scheme (Supplementary Fig. 18). Considering NLT tasks, results show that noise is less detrimental in case of the sine to triangular wave (Supplementary Fig. 16), while the effect of noise in sine to square wave is to smooth transitions between minimum and maximum values of the target output (Supplementary

Fig. 17). However, it should be mentioned that in case of NLT tasks no degradation of the computing performances for $\tilde{g} \rightarrow 0$ and $\tilde{g} \rightarrow 1$ were observed in the deterministic dynamics, since the degradation of the system fading memory properties is expected to affect these computing tasks less.

In this context, it is worth mentioning that a similar time-multiplexing implementation scheme has been reported in a previous simulation work, where NARMA tasks have been implemented by considering dynamics of percolating networks of nanoparticles[50]. Concerning NLT tasks, our implementation based on a single output-node system achieves performances that are comparable to those achieved in simulated multi-output reservoirs based on self-organizing systems[51,52].

## Discussion
Results show that it is possible to model computing systems based on nanoscale networks as continually operating stochastic dynamical systems where signal and information processing can be performed by leveraging the evolution of the physical system. In these nanoscale networks, stimuli-dependent deterministic dynamics and stochastic

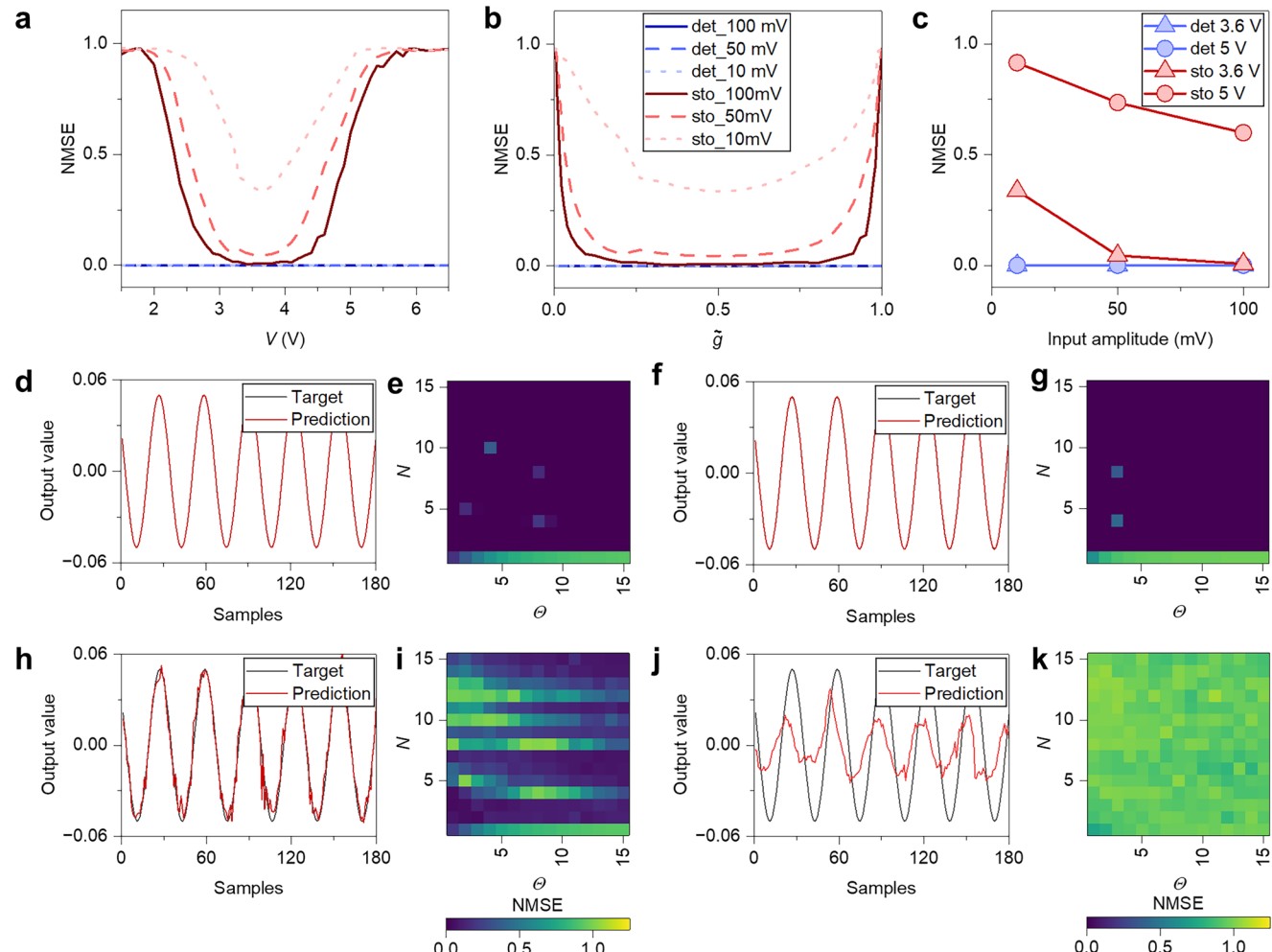

**Fig. 9 | Nonlinear transformation (NLT) task.** Simulation results of the sine to cosine waveform NLT task in terms of NMSE as a function of **a** the bias voltage, and **b** steady state $\tilde{g}$ by considering deterministic and stochastic dynamics and for various amplitudes of the input signal. **c** NMSE as a function of the input signal amplitude for deterministic and stochastic dynamics, by considering polarization voltages of 3.6 V ($\tilde{g}$ - 0.5) and 5 V ($\tilde{g}$ - 0.99). Predictions of the sine to cosine waveform NLT relative to polarization voltages of 3.6 V and 5 V obtained with optimal parameters through deterministic dynamics in **d**, **f**, respectively ([$N, \Theta$] in **d**, **f** are [3,15] and [5,15], respectively). Colormaps showing task performances as a function of $N$ and $\theta$ parameters for polarization voltages of 3.6 V and 5 V in **e**, **g**, respectively. Predictions of the sine to cosine NLT relative to polarization voltages of 3.6 V and 5 V obtained with optimal parameters through stochastic dynamics in **h**, **j**, respectively ([$N, \Theta$] in **h**, **j** are [3,1] and [1,1], respectively). Colormaps showing task performances as a function of $N$ and $\theta$ parameters for polarization voltages of 3.6 V and 5 V in **i**, **k**, respectively. Colormaps and predictions in **d**–**k** refer to results obtained by stimulating the network with a sine wave input with an amplitude of 50 mV.

effects can be holistically modeled as an OU process with jumps, i.e., an Itô-type stochastic differential equation, where the evolutionary state of the system can be probabilistically described by the Fokker-Planck equation with voltage-dependent parameters[53]. The corresponding stationary state of this equation is nothing but the Gaussian distribution obtained in Fig. 4e associated with the minimum of the potential landscape.

It is worth mentioning that the ultimate origin of the modeled network behavior is experimentally difficult to access because of the huge number of junctions involved. Indeed, understanding the origin of deterministic dynamics, noise, and jumps of the conductance time trace necessarily implies high-resolution visualization of the spatially distributed electrical activity across the whole network. Even if it has been shown that it is possible to experimentally investigate the spatial conductance distribution over the network through electrical resistance tomography[29,54], this technique does not allow the spatial resolution required to unveil how switching events in single NW junctions impact the resulting network behavior. Other techniques, such as voltage contrast and resistive contrast scanning electron microscopy

imaging[10,55], besides usually requiring measurements in vacuum that can alter the electrical response of the system, are applicable only to networks with very limited size. Similarly, conductive AFM enables probing the conductance at the single NW junction level but only in small networks consisting of a few NWs[56]. In this context, experiments performed in multiterminal configuration involving synchronous recording of activity in different network areas (such as the recording of voltage maps reported in ref. 19) and the experimental investigation on how power is spatially dissipated through the network (such as through lock-in thermography[11]) can provide further insights on how local activity impacts the resulting network behavior. Details on the state-of-the-art of techniques for the experimental characterization of the network dynamics are reported in Supplementary Note 1.

The deterministic network behavior represents the key aspect for the emulation of synaptic plasticity effects and working memory by exploiting the potentiation and spontaneous relaxation of the conductive pathway connecting stimulated areas[12]. These deterministic processes are fundamental for the implementation of unconventional computing paradigms where information processing occurs by

exploiting the network capability of nonlinearly processing the input signal over time[30]. Understanding and controlling the deterministic dynamics represents a crucial aspect for rational design of the neuromorphic system, allowing for proper identification of the working conditions in terms of operating voltages. In this context, modeling results show the possibility of dynamically setting the steady conduction state through the applied bias voltage and, thus, the operating regime of the self-organizing neuromorphic network. This can enable to control and tailor the system's dynamical regime depending on the specific temporal sequence and amplitude of the input signal, fully exploiting the dynamic range of the system while avoiding operating near the saturation regime. Interestingly, the steady state and dynamics can also be experimentally tailored by controlling the network density (Supplementary Fig. 19). The proposed modeling approach could be integrated into a graph representation, similarly to those reported in refs. 23,57,58, so that the dynamical regime of the system would be tuned depending also by the spatial location of multiple input signals as required for modeling the behavior of multiterminal networks. Additionally, it is worth mentioning that memristive dynamics described by the deterministic form of Eq. (1) have been exploited as dynamics of a physics-inspired recurrent neural network (RNN) computational model[59].

While our modeling approach can well describe the NW networks working in the short-term memory regime exploitable for reservoir computing, we would like to point out that the proposed model can be further extended by introducing (i) transitions between deterministic coexisting states that have been experimentally observed in certain conditions (not modeled here), and (ii) long-lasting variations in the steady states to emulate also long-term changes in the network conductance that can be experimentally observed under specific stimulation conditions[29] (for these purposes, the introduction of new parameters able to account for drifts of the steady states are required).

In the framework of reservoir computing, the reported approach allows to quantify the influence of deterministic and stochastic dynamics on computing capabilities. While the reported model, based on a combination of physical foundations and mathematical properties and able to describe the behavior of the system in terms of a minimal number of assumptions, can be adopted to explore in silico different implementations of the reservoir computing paradigm through simulations, further work is required to experimentally delineate the computing capabilities of these systems. In this context, it is worth noticing that reservoir computing has been experimentally explored in a wide range of self-organizing memristive systems, including nanotubes[60], nanoparticles[61,62], and nanowires[19,20,31], by exploiting multiterminal configurations that allow to probe the evolution of the internal state of the system as seen from different spatial locations.

Simulation results show that stochastic effects limit the separability property of the system (i.e., the capability of the system to differentiate its response when processing different input signals). When little or no memory is required for the completion of the task (such as in the case of NLT), computing capabilities of the system rely mainly on the deterministic-to-stochastic (signal-to-noise) response of the network. When memory is essential for the completion of the task (such as in NARMA), computing capabilities rely also on the dynamical regime of the network and its fading memory capabilities determined by the steady state. In both cases, optimization of the computing capabilities relies on the proper selection of the input signal amplitude. For instance, operating the network at $\tilde{g} \sim 0.5$ enhances the system's dynamical range. Additionally, the results show that stochastic effects should be properly considered for the optimization of the time-multiplexed reservoir computing implementation, for the proper selection not only of $[N, \Theta]$ parameters but also for the design of an appropriate masking scheme for the input signal. As a result, the masking scheme needs to be optimized to reduce degradation of

computing performances for a given set of $[N, \Theta]$ by means of, for example, the use of supervised learning algorithms. More in detail, it is important to point out that stochastic effects do not hinder the possibility of operating the system with a low number of virtual nodes $N$, as required for reducing the hardware complexity of the system. Indeed, a lower $N$ number allows the system to operate with a reduced number of weights to be trained (and a lower amount of information to be temporally stored before being analyzed by the readout).

Even if at first sight the random nature of the output signal seems to be detrimental for computing when adopting a deterministic perspective, it was shown that stochastic dynamics can indeed be exploited as an additional dimension for the implementation of stochastic learning rules. This comprises the hardware realization of random number generators, physical unclonable functions, and chaotic/stochastic computing systems by taking advantage of the material substrate as the underlying source of randomness[63].

In all these contexts, Eq. (1) can be exploited as an electrical transfer function to model the system's dynamical output corresponding to arbitrary input signals for a rational design of neuromorphic systems based on self-organizing nanoscale networks. This can represent a step ahead for the conceptualization of a general theory of computing with non-linear dynamics and stochastic effects through a dynamical system-oriented view[27,64,65]. Furthermore, we envision that dynamics of complex systems, like the one analyzed in this work, can be exploited for *in materia* forecasting the evolution of stochastic variables that can be generically represented by means of Orstein-Uhlenbeck processes, such as financial processes including the evolution of interest rates describable through the Vasicek or Hull-White models[66]. Indeed, after proper calibration, the inherent capability of the physical system to approximate the dynamic evolution of the stochastic differential equations describing specific processes (in particular OU process) could enable forecasting the evolution of stochastic trajectories by observing, given an initial condition, how the physical observables evolve over time. More in generally, these results can pave the way for the development of alternative concepts of unconventional computing paradigms that take advantage of both deterministic and stochastic dynamics on the same physical substrate, as naturally occurs in biological systems.

In summary, we showed that neuromorphic nanowire networks can be modeled as a stochastic dynamic system. We show that deterministic and stochastic dynamics of the physical system can be qualitatively and quantitatively described in a unified mathematical framework such as the Ornstein-Uhlenbeck process. This approach enables to holistically describe stimuli-dependent deterministic trajectories of the system as well as conductance fluctuations and jumps. Furthermore, the proposed compact model description can be exploited to quantitatively assess the impact of deterministic and stochastic dynamics on computing capabilities of these physical systems in the framework of reservoir computing, as shown by considering benchmark tasks. These results can pave the way for the rational and optimized development of neuromorphic systems that can fully exploit their spatio-temporal dynamics similarly to our brain.

## Methods
### Fabrication of self-organizing neuromorphic networks
Self-organizing Ag NW networks have been fabricated by means of drop-casting[12], by dispersing Ag NWs with length of 20–50 μm and a diameter of ~115 nm in isopropyl suspension (from Sigma–Aldrich) on an insulating $SiO_2$ substrate. Details on structural and chemical characterization of Ag NWs are analyzed in our previous work[12]. The fabrication of neuromorphic NW networks consisted in drop casting Ag NWs in isopropanol solution (drop of 20 μL, concentration of ~0.09 mg mL$^{-1}$) on a ~1 × 1 cm$^2$ substrate, followed by deposition of Au metal electrodes through sputtering deposition and shadow mask. To show the dependence of the steady state on the network density

(Supplementary Fig. 15), results have been compared with the response of a different NW network with higher density realized by drop casting Ag NWs in isopropanol solution (drop of 20 μL) with a concentration of ~0.13 mg mL⁻¹. The morphology of the self-organizing NW network and details of NW junctions were assessed by means of Scanning Electron Microscopy (SEM, FEI Inspect F).

## Experimental characterization

Experimental electrical characterization has been performed at a controlled constant temperature of 303 K in a hermetically closed environment of ambient air, by contacting sample substrate with a thermocouple controlled through a Lake Shore 331 temperature controller. Electric measurements have been carried out by a Keithley 6430, connected through a preamplifier in two-terminal fashion to facing electrodes sputtered at sample edges centers (electrode distance of ~7 mm). Current has been recorded in auto range mode at fixed minimum range of 1 nA and with Number of Power Line Cycles (NPLC) set at 1, where voltage was sourced at 20 V range. Under these conditions, the sampling rate was ~1.6 Hz. Measurements were acquired in pulse-train fashion with 10 h width pulses of voltage progressively increasing from 0.1 V to 6.6 V with 0.1 V step, separated by 10 mV reading intervals of 1 h that ensure network relaxation to the ground state before stimulation with the subsequent voltage amplitude. The mean value of stationary state $\tilde{G}$ reported in Fig. 2b was evaluated for each voltage bias condition by considering the signal after ~18,500 s from bias application to avoid transient dynamics, averaging the stationary signal over ~15,000 s.

The disentanglement between Gaussian noise and jump events in the $dG/dt$ signal has been performed by determining the interval underlying the most Gaussian data region in a Kolmogorov-Smirnov (KS) sense. In detail, the $p$ values of progressively larger data intervals (centered in 0) have been evaluated by using the KS test. The extreme ensuring maximum $p$ value, i.e., most probable Gaussian distribution, has been chosen as the threshold between noise and jump events.

The probability distributions of inter-event intervals $p$(IEI) as a function of IEI reported in Figs. 3h and 4h have been obtained by linear binning of data with size of 10 s. Instead, the probability distributions of jumps $p(\Gamma)$ as a function of jump amplitude $\Gamma$ reported in Figs. 3i and 4i have been obtained by logarithmically binning data with a density of 50 bins per decade. Power-law fittings of data reported in Figs. 3i and 4i have been performed by following procedure used for complexity analysis in ref. 67. A doubly truncated target distribution has been considered, where the lower cutoff emerges as a consequence of instrumental finite resolution, while the upper one results from the finite measurement duration and sampling. The data fraction to be fitted has been progressively shrunk and the related power-law exponent has been extracted by means of maximum likelihood estimation (MLE). The exponent has been then validated by using KS test in the following way: 500 test power-law distributions have been generated starting from the exponent under study, and their KS statistics with respect to fitting line (i.e., the maximum absolute difference between their respective cumulative distribution functions) has been evaluated. KS statistics have also been evaluated between the experimental data and the fit. The fitting has been considered acceptable if the experimental KS statistics have resulted lower than the test ones in the 20% of the cases at least. The validated fitting obtained from the least truncated data interval has been then selected as the final one. Exponent uncertainty has been obtained through the Monte Carlo method by fitting the 500 test power-law distributions used for validation and evaluating their exponent standard deviation. The experimental number of events $n$ as a function of time was interpolated to extract the event rate with a straight line with intercept 0 (i.e., 0 events at $t = 0$) to obtain the experimental event rate $\lambda$.

## Modeling

Modeling was performed in Python. The deterministic balance-rate equation reported in Eq. (3) can be solved analytically, where the recursive (iterative) solution can be expressed as (assuming a simulation timestep $\Delta t > 0$ and knowing the initial value $g_0$)[44]:

$$g_t = \tilde{g}\left(1 - e^{-\theta \Delta t}\right) + g_{t-1}e^{-\theta \Delta t} \tag{7}$$

As an alternative, Eq. (3) can be solved by using the Euler method as a first-order numerical procedure for solving ordinary differential equations, expressing the solution as:

$$g_t = g_{t-1} + \theta\left[\tilde{g} - g_{t-1}\right]\Delta t \tag{8}$$

Parameters for modeling the deterministic behavior of the network were retrieved from interpolation of the stationary state conductance value $\tilde{G}$ over applied voltage biases reported in Fig. 2b.

The stochastic differential equation describing the Ornstein-Uhlenbeck process reported in Eq. (1) can be solved analytically in case of no jumps ($\Gamma = 0$), where the recursive (iterative) solution can be expressed as:

$$g_t = \tilde{g}\left(1 - e^{-\theta \Delta t}\right) + g_{t-1}e^{-\theta \Delta t} + \varepsilon_t \tag{9}$$

Where $\varepsilon_t$ is normally distributed with mean zero and standard deviation $\sigma_e$ is:

$$\sigma_e{}^2 = \left[1 - e^{-2\theta}\right]\frac{\sigma^2}{2\theta} \tag{10}$$

In case of jumps ($\Gamma \neq 0$), the stochastic differential equation can be numerically solved through the Euler−Maruyama method, where the solution can be expressed as:

$$g_t = g_{t-1} + \theta\left[\tilde{g} - g\right]\Delta t + \sigma\sqrt{\Delta t}\xi + \Gamma \Delta q, \ \xi \sim N(0,1) \tag{11}$$

where $\xi$ is a random Gaussian variable with variance 1 (independent at each time step). The normalization factor $\sqrt{\Delta t}$ comes from the fact that the infinitesimal step for a Brownian motion has the standard deviation $\sqrt{\Delta t}$. The stochastic model was calibrated on experimental data: (i) by selecting the $\sigma$ value that gives rise to the standard deviation of the Gaussian noise distribution that matches the experimental one, and (ii) by assigning to jump events a jump amplitude sampled from a bounded power law distribution with exponent $\alpha$ that matches experimental results and bounded values that match the minimum and maximum experimental value ($\Gamma_{min}$ and $\Gamma_{max}$), where the minimum value matches with the experimental threshold for noise disentangle (i.e., modeled jump events have amplitude larger than the threshold value for noise disentangle). The jump direction during modeling is randomly assigned. While the probability of jump direction is the same when the device is in the stationary state, the probability of jump direction is assumed to be proportional to $(\tilde{g} - g)$. Where the jump up probability is $0.5 + A^*(\tilde{g} - g)$ and jump down probability is $0.5 - A^*(\tilde{g} - g)$ ($A^*$ is a normalization constant that depends on the maximum jump amplitude to normalize probability to 1).

The potential function $U = -\int \theta\left[\tilde{g} - g\right]dg = \theta g\left(\frac{g}{2} - \tilde{g}\right) + C$, where $C$ is an arbitrary constant set to 0 in our work, was obtained by integrating $dg/dt = -\partial U/\partial g$, where the deterministic form of Eq. (1) was considered as $dg/dt$. Information processing capabilities in Fig. 7 have been analyzed by considering modeling parameters $\theta(V)$, $\tilde{g}(V)$ extracted from the experimental data reported in Fig. 2, and $\sigma, dW, \Gamma, dq$ extracted from the experimental data reported in Fig. 3.

## Time-multiplexed reservoir computing implementation

Benchmarking of computing capabilities of the deterministic and stochastic network model has been performed by implementing a time-multiplexed reservoir computing scheme[47], exploiting the network in two-terminal configuration as a single dynamical node with delayed feedback. In this implementation, the transient response to a masked input signal of the network in two terminal configuration is sampled $N$ times (virtual nodes) with separation time $\Theta$ during each timestep of the input signal (details on the implementation in Supplementary Fig. 11). Results reported in the manuscript have been obtained by adopting the masking scheme reported in Supplementary Fig. 15 panel a, if not differently specified. A similar implementation was also reported in percolating networks of nanoparticles[50].

## Training

The readout function (output layer) was trained through a supervised learning algorithm. For a $N$-dimensional reservoir output $\mathbf{X} = [x_1, x_2, \ldots, x_N]$, training involved the calculation of a vector of linear coefficients $\mathbf{w} = [w_1, w_2, \ldots, x_N]$ such that the predicted output of the system $\hat{y}(k)$ well approximate the target output $y(k)$:

$$\hat{y}(k) = \sum_{i=1}^{N+1} w_i x_i(k) \approx y(k) \tag{12}$$

Linear coefficients were calculated through linear regression.

## Nonlinear autoregressive moving average task

NARMA tasks are a set of time series prediction tasks involving the emulation of nonlinear dynamical systems, representing a challenging task for computational systems because of their nonlinearity and dependence on previous time lags. The task involves learning the association between a discrete input white noise $u(k)$ and the chaotic time series generated by the nonlinear dynamical system. For the second-order NARMA system (NARMA-2), the time series is given by:

$$y_{k+1} = \alpha y_k + \beta y_k y_{k-1} + \gamma u_k^3 + \delta \tag{13}$$

with $\alpha = 0.4$, $\beta = 0.4$, $\gamma = 0.6$, $\delta = 0.1$. The white input noise was fed to the network after being scaled by the desired amplitude (amplitudes of 10, 50, and 100 mV were considered). 720 timesteps of the system outputs for were used for training the readout function, while testing was performed on 180 timesteps. Performances of the system were evaluated through NMSE. In Fig. 8a, b, for each polarization voltage bias (for each $\tilde{g}$), the NMSE represents the best NMSE obtained by evaluating computing performances for $N$ and $\Theta$ parameters in the range [0, 15]. In other words, this represents the NMSE optimized in terms of $N$ and $\Theta$ (in the selected parameter ranges), i.e., the optimized performance of the system given a $\tilde{g}$.

## Nonlinear transformation task

NLT tasks are a set of tasks involving a nonlinear transformation of a sine wave input signal[49]. This task requires mainly a high degree of nonlinearity. We considered the transformation of the input signal into a cosine, square, or triangular waveform of the same period. For this purpose, the sine wave input signal is fed to the network after being scaled by the desired amplitude (amplitudes of 10, 50, and 100 mV were considered). Eight hundred timesteps of the system outputs were used for training the readout function, while testing was performed on 200 timesteps. Performances of the system were evaluated through NMSE. In Fig. 9a, b, for each polarization voltage bias (for each $\tilde{g}$), the NMSE represents the best NMSE obtained by evaluating computing performances for $N$ and $\Theta$ parameters in the range [0, 15]. In other words, this represents the NMSE optimized in terms of $N$ and $\Theta$ (in the selected parameter ranges), i.e., the optimized performance of the system given a $\tilde{g}$.

## Normalized mean square error

Task performances were evaluated by the normalized mean square error (NMSE) between predicted output $\hat{y}(k)$ and target output $y(k)$, through the equation:

$$NMSE = \frac{1}{K} \frac{\sum_{1}^{K} (\hat{y}_k - y_k)^2}{\sigma^2(y_k)} \tag{14}$$

where the sum of square residuals is normalized by the variance of the target function $\sigma^2(y_k)$.

## Data availability

The data that support the findings of this study are available on Zenodo (https://doi.org/10.5281/zenodo.15050217). All other data are available from the authors.

## Code availability

The codes used to generate datasets of simulations can be accessed on GitHub (https://github.com/MilanoGianluca/Self-organizing_neuro morphic_networks_as_stochastic_dynamical_systems). The code release is available on Zenodo (https://doi.org/10.5281/zenodo.15174744).

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

## Acknowledgements

G.M. acknowledges support by the European Research Council (ERC) under the European Union's ERC Starting Grant (ERC-2024-STG) agreement "MEMBRAIN" No. 101160604. Part of this work was supported by the European project MEMQuD, code 20FUN06. This project (EMPIR 20FUN06 MEMQuD) has received funding from the EMPIR program co-financed by the Participating States and from the European Union's Horizon 2020 research and innovation program. Part of this work has been supported by NEURONE, a project funded by the European Union— Next Generation EU, M4C1 CUP No. I53D23003600006, under program PRIN 2022 (prj code 20229JRTZA). Part of this work has been carried out at Nanofacility Piemonte INRiM, a laboratory supported by the "Compagnia di San Paolo" Foundation, and at the QR Laboratories, INRiM. E.M. acknowledges project PID2022-139586NB-C41 funded by MCIN/AEI/ 10.13039/501100011033, Spain, and FEDER, E.U.

## Author contributions

G.M. and E.M. generated the idea, designed the experiments, developed modeling, analyzed data, and wrote the manuscript. F.M. performed experimental activities and supported data analysis. D.P. supported modeling activities and data analysis. G.M., F.M., D.P., C.R., and E.M. participated in the discussion of results and revision of the manuscript.

## Competing interests

The authors declare no competing interests.
