## [Transparent Peer Review file · Nature Communications]

Self-organizing neuromorphic nanowire networks as stochastic dynamical systems

Corresponding Author: Dr Gianluca Milano

Version 0:

Reviewer comments:

Reviewer #1

(Remarks to the Author)

The paper reports that the stochastic and deterministic behavior of time-dependent conductance of the voltage-stimulated Ag nanowire network can be described by Ornstein-Uhlenbeck (OU) process by assuming appropriate jumps of conductance. I believe the work certainly provide scientifically important insights to understand stochastic dynamics emerging in the complex network. Increasing interest in the feasibility of computing with undesigned conducting/complex networks, and owing to the increasing demands to understand and achieve reasonable- and systematic-control of the electrical properties of complex network such as that studied by the authors, appropriate formulation to describe stochastic behavior is very important. Thinking of the importance of the author's work, I recommend the paper be accepted for publication in Nature Communications after clarifying the following point.

Although the paper is very nicely organized and providing important message, I do have one question to better understand the meaning of the formulation with OU scheme with jumps. As the authors pointed out, there are noises and jumps in the signal, which has been reported in previous studies by other groups. For example, the paper by Diaz-Alvarez et al.(ref.18) also pointed out that (a sort of) jump induces transition of the whole network from a metastable state to another metastable state. When such a situation is formulated by Equation (1) in the paper, noises and jumps are clearly separated while the authors say that both are a consequent of random/stochastic behavior of switches (nanowire-nanowire junctions). Actually, what's a difference between noise and jump?

Is a jump coming from accidental coincidence of ON-OFF events of multiple switches? Or is the jump suggesting formation of domains in the network which may results in occurrence of domain-switching with only one or a few switches? If the authors visualize the conductance (or resistance) distribution over the network, this point will become clear, I guess.

Reviewer #2

(Remarks to the Author)

In the manuscript entitled "Self-organizing neuromorphic nanowire networks are stochastic dynamical systems", the authors present a modelling method for the self-organizing nanowire network. It is interesting to see the complex dynamics, including deterministic and stochastic behaviors, can be unified by the statistics method based on Ornstein-Uhlenbeck process, which could provide useful insights into the nanowire networks for potential applications in computing. However, there are several critical concerns to be addressed before it can be reconsidered for publication.

1) The main problem is the incompleteness of this study. The authors have done a good work in using new method for characterization and modelling, while mentioning multiple times "the relationship between emergent dynamics and information processing capabilities...", "for rational development of physical computing...", "results can pave the way for the development of neuromorphic system...". However, there is no connection between the proposed method and any neuromorphic system or information processing capabilities, resulting in a gap between the results and the points that the authors intend to claim. The authors should further showcase 1) what is the target information processing method or neural network? 2) how this method can guide and optimize a "rational" neuromorphic system design? 3) how to analytically and quantitatively connect the deterministic and stochastic results with hardware performance? 4) what are the performance matrices that can be improved by the so-called "rational design"? e.g., accuracy in benchmark tasks, computing efficiency

and implementation cost.

- 2) Furthermore, it is also hard to find a reasonable relation between the attractor states and information processing in the "Relationship between attractor states and information processing" section. There is neither quantification of information processing capacity, nor processing results for the triangular signal. The last conclusive sentence in this section "these results show ... of the network" is problematic and doesn't technically sound. Why the fixed-point attractor of the network can enhance the information processing capabilities? What kind of information processing is it? This must be clarified.
- 3) Following the last two comments, the next unclear point is how to properly control the attractor and thus control the information processing capabilities? Naively thinking, it is hardly controllable once the nanowire network is fabricated.
- 4) In Fig. 7, are the results from simulations or experiments? The authors just mentioned "observe". Please clarify. If these are experiment, how to decouple the deterministic and stochastic signals?
- 5) In Fig. 7, can the authors compare the modelling and experiment results under triangular inputs?
- 6) The authors use the proposed method to model a single-input-single-output nanowire network. However, it might operate in a multiple-input-multiple-output fashion when the authors previously used the same material for reservoir computing, e.g. ref. 25. On one hand, there is no target computing paradigm in this manuscript, as discussed in comments #1 and #2. On the other hand, is the SISO model in this paper applicable for scaling up to describe the potential computing methods?
- 7) Figs. 3 and 4 could be combined into one figure to more clearly compare the modelling performance.
- 8) As a modelling work, it is recommended to provide the simulation model that simultaneously describe deterministic and stochastic processes to facilitate the followers to understand the details.

Reviewer #3

(Remarks to the Author)
See attached file

Version 1:

Reviewer comments:

Reviewer #1

(Remarks to the Author)

The paper reporting the stochastic and deterministic behaviour of the time-dependent conductance of the voltage-stimulated Ag nanowire network with an interpretation using the Ornstein-Uhlenbeck (OU) process will certainly provide insights towards better understanding and achieving controlled exploitation of complex networks. Due to the complexity of the material system, it's usually difficult to understand the whole behaviour of such a system (complex network) without computation under certain conditions, but the authors have shown the possibility to formulate the behaviour with a minimal number of assumptions.

Although there are still open questions (probably difficult to solve at the moment), I believe that the paper is worthwhile to be published in its present form, which would result in stimulating related research towards future computation. And I expect that the work would be cited as one of the milestones in the course of developing neuromorphic computing that does not consume energy as has been developed so far within the framework of current computing schemes based on von Neumann architecture.

(Remarks on code availability)

I do not have sufficient background to review the code itself. Sorry.

Reviewer #2

(Remarks to the Author)

Substantial improvements have been made in the revised manuscript, especially the additional simulation for demonstrating the performance in reservoir computing. There is one remaining concern before it can be considered for publication in Nature communications.

In the authors' response R3(3), the authors mentioned "3) We quantitatively connected deterministic and stochastic dynamics with computing performances". By learning Fig. 7-9 and the relevant text, I feel like the simple fact is that the deterministic evolution of the memory state provides much less noisy dynamics than the stochastic evolution (Fig. 7c-f), resulting in a lower NMSE (Fig. 8c and 9c). Meanwhile the rest figures regarding bias, theta and N, are more like an optimization strategy, as reservoir computing is known to be highly sensitive to those parameters. The way that the authors connect the det&sto with computing performances is simply tuning the relevant parameters. It can only be concluded from these results that a less noisy system yields a lower NMSE. The authors should give more in-depth insights on the effect of det&sto and the underlying mechanism.

(Remarks on code availability)

Reviewer #3

(Remarks to the Author)
See attached file.

(Remarks on code availability)

Version 2:

Reviewer comments:

Reviewer #2

(Remarks to the Author)

While the random nanowire network has its own flaws for reservoir computing, it is worthy investigating the so-called self-organizing capability. Other than that, I do not have any further comments.

(Remarks on code availability)

I do not have sufficient background to review the code itself

Reviewer #3

(Remarks to the Author)

The authors have written a long response and made some further revisions to the manuscript, but they have not addressed the concerns I have raised. The authors repeatedly state that the emergent behavior and related effects of interest are evidenced by the results of their model. The model (eq 1) clearly includes the coupling between the effects of interest. But, as I emphasized in my previous reports, there is no evidence in the experimental data for coupling of these effects, and that is still the case in the revised manuscript.

The fact that a model that includes coupling can be used to generate effects similar to those in the experiments does not mean that the coupling is present in the experiments. A far simpler and more obvious explanation of the observed effects is that they occur through independent processes e.g. dynamical effects are observed in the conductance versus time and autocorrelation plots but this is not evidence for intertwined deterministic and stochastic dynamics as is claimed.

Comments on some of the changes to the manuscript:

- every system that has interactions between components is not emergent
- the demonstration in figure S2 that the conductance of the devices reach the same value each time 3V is applied does not convince me that there is an attractor - similar behavior could result from quite simple circuits and in this system would arise simply from sequential formation of a chain filaments.
- The new Fig S5(a) is the only evidence I see for mean reversion in experiments, but even there the jumps appear atypical (no data like that the right panel is visible in the left panel) and the mean values in the two panels are different.
- It is clear now that the reservoir computing aspect of the paper is done through simulations but the fact that simulations have to be used amplifies the concerns I have about the lack of coupling / complex dynamics in the experimental data.

I can however see that the claimed coupling potentially might eventually be observable in nanowire networks and so I want to suggest a route toward publication. I wonder if the authors would refocus their paper on results from simulations and include a section that describes adaptations of the experiments that would allow emergent behavior to be observed. This would potentially avoid the issues in relation to the experimental data that I believe are misleading at present.

(Remarks on code availability)

Version 3:

Reviewer comments:

Reviewer #2

(Remarks to the Author)

In the revised manuscript, the authors made improvement, mainly on the previously misleading terms and concepts, to address reviewer 3's comments. I agree with those changes and suggest this latest version to be published.

(Remarks on code availability)

Reviewer #3

(Remarks to the Author)

See attachment

(Remarks on code availability)

Point-by-point response

We thank the Reviewers for the constructive and competent comments, which give us the opportunity to improve the overall quality of our manuscript. We provide here a point-by-point response addressing all the reviewer comments supported by new experimental and simulation results. We have extensively revised the main manuscript and added supplementary information, accordingly. We believe that the revised version of our manuscript now complies with the high standard required for publication in *Nature Communications*.

Reviewer #1 (Remarks to the Author):

The paper reports that the stochastic and deterministic behavior of time-dependent conductance of the voltage-stimulated Ag nanowire network can be described by Ornstein-Uhlenbeck (OU) process by assuming appropriate jumps of conductance. I believe the work certainly provide scientifically important insights to understand stochastic dynamics emerging in the complex network. Increasing interest in the feasibility of computing with undesigned conducting/complex networks, and owing to the increasing demands to understand and achieve reasonable- and systematic-control of the electrical properties of complex network such as that studied by the authors, appropriate formulation to describe stochastic behavior is very important. Thinking of the importance of the author's work, I recommend the paper be accepted for publication in *Nature Communications* after clarifying the following point.

We thank the reviewer for the general positive assessment on the proposed formulation to describe the stochastic behavior exhibited by our devices, and for finding the manuscript worthwhile to consider publication in *Nature Communications*. We thank for the constructive comments that gave us the possibility to improve our work.

C1. Although the paper is very nicely organized and providing important message, I do have one question to better understand the meaning of the formulation with OU scheme with jumps. As the authors pointed out, there are noises and jumps in the signal, which has been reported in previous studies by other groups. For example, the paper by Diaz-Alvarez et al.(ref.18) also pointed out that (a sort of) jump induces transition of the whole network from a metastable state to another metastable state. When such a situation is formulated by Equation (1) in the paper, noises and jumps are clearly separated while the authors say that both are a consequent of random/stochastic behavior of switches (nanowire-nanowire junctions). Actually, what's a difference between noise and jump?

R1. We thank the reviewer for pointing out this aspect that was not appropriately discussed in the manuscript. While noise relies on stochastic effects occurring at nanowire-nanowire junctions from intrinsically probabilistic atomic-scale switching processes, jumps can be interpreted as a manifestation of the network transitions between metastable states due to the connection/disconnection of domains in the network. As analyzed in ref.¹, current jumps (or conductance jumps) can be attributed to resistive switching events in vital topological areas of the network that electrically connect/disconnect network domains. In this context, we can describe jumps as special cases of fluctuations. Beyond the magnitude of the fluctuation, there is also a difference between noise and jumps in the temporal scale. Indeed, jumps are associated with “rare” events characterized by an intensity (number of events per unit of time).

C2. Is a jump coming from accidental coincidence of ON-OFF events of multiple switches? Or is the jump suggesting formation of domains in the network which may results in occurrence of domain-switching with only one or a few switches? If the authors visualize the conductance (or resistance) distribution over the network, this point will become clear, I guess.

R2. As discussed in response R1, jumps can be attributed to transitions of the network from one metastable state to another. This involves the connection/disconnection of domains in the network through the switching of one (or few) junctions that are localized in specific topological areas of the network. In the framework of the description of the network as a complex system through graph theory, conductance jumps can be described as related to switching events of junctions characterized by high centrality, i.e., to the change of the resistive state of those junctions that, due to their peculiar location on the network topology, play a more relevant role in regulating the current flow through the system. The influence of high centrality junctions on the transmission of information (i.e., on current flow) in nanowire networks have been analyzed through simulations in ref.². Consequently, the stochastic switching of one (or few) of junctions with high centrality can be associated to the electrical connection/disconnection of network domains associated with large changes in the electrical conductivity of the system manifested as jumps in the conductance time trace.

Concerning the experimental visualization of the conductance over the network, this represents a challenge. Even if we showed in our previous work that it is possible to experimentally investigate the spatial conductance distribution over the network through electrical resistance tomography³, this technique does not allow the spatial resolution to observe the electrical behavior of single NW junctions as required for the direct observation of jumps. Other techniques such as the voltage contrast SEM imaging⁴ require making measurements in vacuum, altering electrochemical effects governing resistive switching events in NW junctions and, thus, altering the emergent behavior and electrical response of the system. In this context, jumps observed in our work can be explained according to the discussion based on simulations reported in the work by Diaz Alvarez et al. in ref. ².

Changes in the manuscript: The aspects discussed in response R1 and R2 were clarified in the revised manuscript, at page 6, clarifying the difference between noise and jumps, with reference to the work of Diaz-Alvarez et al. ¹.

Reviewer #2 (Remarks to the Author):

In the manuscript entitled “Self-organizing neuromorphic nanowire networks are stochastic dynamical systems”, the authors present a modelling method for the self-organizing nanowire network. It is interesting to see the complex dynamics, including deterministic and stochastic behaviors, can be unified by the statistics method based on Ornstein-Uhlenbeck process, which could provide useful insights into the nanowire networks for potential applications in computing. However, there are several critical concerns to be addressed before it can be reconsidered for publication.

We thank the reviewer for the general positive assessment on our work and for the constructive comments that gave us the possibility to improve our work.

C3. The main problem is the incompleteness of this study. The authors have done a good work in using new method for characterization and modelling, while mentioning multiple times “the relationship between emergent dynamics and information processing capabilities...”, “for rational development of physical computing...”, “results can pave the way for the development of neuromorphic system...”. However, there is no connection between the proposed method and any neuromorphic system or information processing capabilities, resulting in a gap between the results and the points that the authors intend to claim. The authors should further showcase 1) what is the target information processing method or neural network? 2) how this method can guide and optimize a “rational” neuromorphic system design? 3) how to analytically and quantitatively connect the deterministic and stochastic results with hardware performance? 4) what are the performance matrices that can be improved by the so-called “rational design”? e.g., accuracy in benchmark tasks, computing efficiency and implementation cost.

R3. Following the reviewer’s recommendations, we have extended our work to connect the proposed description of the network as a stochastic dynamical system and its information processing capabilities in the framework of reservoir computing.

With reference to the points raised by the reviewer, we have very briefly summarized new results included in the revised version of the manuscript:

- 1) **We have established a connection between the proposed model and reservoir computing**, implementing this computing paradigm through a time-multiplexed scheme by using a single dynamical node with delayed feedback
- 2) **We have shown that this modeling approach allows to investigate and disentangle the effect of deterministic and stochastic dynamics on computing capabilities of the system**, as shown by testing computing capabilities on standard benchmark computing tasks, namely the time series prediction Nonlinear autoregressive moving average (NARMA) task⁵ requiring both nonlinearity and memory, and Nonlinear Transformation tasks (NLT)⁶ including sine to cosine, sine to square and sine to triangular transformations.
- 3) **We quantitatively connected deterministic and stochastic dynamics with computing performances**, as shown by analyzing the dependence of the normalized mean-square error

(NMSE) through deterministic and stochastic dynamics of the system on computing tasks requiring both nonlinearity and memory capabilities.

- 4) **We show that performance in terms of NMSE in benchmark tasks can be improved and optimized by properly selecting the network operating regime** in terms of attractor state (i.e., by controlling the bias voltage) and the amplitude of the input signal. In this way, we show that it is possible not only to regulate the deterministic-to-stochastic response but also its fading memory capabilities.

In this context, the modeling platform can be exploited not only for the design of nanowire-based computing systems in the framework of reservoir computing but also for testing new concepts of computing implementations leveraging the coexistence of deterministic and stochastic effects on the same physical substrate.

Changes in the manuscript: new results are reported Figures 8 and 9 as well as in new Supplementary Figures S13 and S14 and discussed in detail at pages 23-26 of the revised manuscript. All details on the computing implementation are reported in new methods sections and in new Supplementary Figure S12. All these aspects were clarified throughout the whole manuscript, including abstract, introduction, discussion and conclusions.

C4. Furthermore, it is also hard to find a reasonable relation between the attractor states and information processing in the “Relationship between attractor states and information processing” section. There is neither quantification of information processing capacity, nor processing results for the triangular signal. The last conclusive sentence in this section “these results show ... of the network” is problematic and doesn’t technically sound. Why the fixed-point attractor of the network can enhance the information processing capabilities? What kind of information processing is it? This must be clarified.

R4. We agree with the reviewer that the title of the paragraph “*Relationship between attractor states and information processing*” can be somewhat misleading for the reader. For this purpose, we have clarified the relationship between attractor states, information processing, and computational capabilities of the system by providing new results on the implementation of standard benchmark computing tasks in the framework of reservoir computing, as detailed in response R3.

Changes in the manuscript:

- The old paragraph “*Relationship between attractor states and information processing*” was changed by “*Relationship between attractor states and signal processing*” in order to avoid any misunderstanding. The text of this paragraph was also improved.
- The conclusive sentence of the above-mentioned paragraph was rewritten according to the reviewer’s suggestion.
- New results reporting the quantification of information processing capabilities (by processing network outputs) were reported in the framework of reservoir computing. New results have been reported in the new Figures 8 and 9, as well as in new Supplementary Figures S13 and S14.

C5. Following the last two comments, the next unclear point is how to properly control the attractor

and thus control the information processing capabilities? Naively thinking, it is hardly controllable once the nanowire network is fabricated.

R5. The attractor state of the network can be controlled through the bias voltage, as shown by experimental and modeling results reported in Figure 2b. Here, the experimental attractor state of the system in terms of conductance (i.e., the network conductance state after transients) is reported versus the applied voltage. As can be observed, the attractor state of the system can be tuned in between a minimum and a maximum value by properly controlling the applied voltage.

While the bias voltage allows to tune the attractor state of the system in the same NW network, we provide new results showing that the attractor state of the system can be tuned also by changing the network topology, i.e., by changing the network density. New results are reported in Supplementary Figure S15 where the conductance time trace under stimulation with the same bias voltage applied to different networks characterized by different densities are compared. Even if the effect of network density on network dynamics is out of the scope of this work, the results are highly promising since they show that the same voltage bias yield different attractor states. Preliminary data shows that, besides resulting in different transient dynamics (i.e., in different reversion speed θ), a higher density network results in a higher conductive attractor state.

Changes in the manuscript:

- New experimental results showing the possibility of tuning the attractor state of the network depending on the network topology are reported in the new Supplementary Figure S15.
- It was clarified through the manuscript (discussion paragraph, page 27-28) that attractor states can be further tuned by controlling the network density, with reference to the new results reported in Supplementary Figure S15.

C6. In Fig. 7, are the results from simulations or experiments? The authors just mentioned “observe”. Please clarify. If these are experiment, how to decouple the deterministic and stochastic signals?

R6. Results reported in Figure 7 are from simulations since the scope of this figure was to show that the model allows to disentangle the effect of deterministic and stochastic dynamics on the evolution of the internal memory state of the network (this disentanglement cannot be done with experimental data).

Changes in the manuscript: it was clarified that the aim of Figure 7 is to show through modeling the possibility of disentangling deterministic and stochastic dynamics on the evolution of the internal memory state of the network (page 20 of the revised manuscript). This aspect was further clarified by clearly specifying in the figure caption that results of this figure are from simulations.

C7. In Fig. 7, can the authors compare the modelling and experiment results under triangular inputs?

R7. As detailed in response R6, the aim of this figure is to show through modeling interpolated on experimental data the possibility of disentangling deterministic and stochastic dynamics on the evolution of the internal memory state of the network.

Changes in the manuscript: we have clarified through the revised manuscript the aim of modeling results reported in Figure 7 (page 20).

C8. The authors use the proposed method to model a single-input-single-output nanowire network. However, it might operate in a multiple-input-multiple-output fashion when the authors previously

used the same material for reservoir computing, e.g. ref. 25. On one hand, there is no target computing paradigm in this manuscript, as discussed in comments #1 and #2. On the other hand, is the SISO model in this paper applicable for scaling up to describe the potential computing methods?

R8. The here proposed method to model was exploited to model the response of a single-input-single-output nanowire network. While in previous works we proposed an implementation of reservoir computing with multiple input/outputs (refer to ref. ^{7,8}), here reported new results show that it is possible to implement reservoir computing also in two-terminal nanowire networks by exploiting a time-multiplexed reservoir computing scheme by using a single dynamical node with delayed feedback (refer to the work of Appeltant et al.⁹ for a theoretical description of this reservoir computing implementation). Details can be found in response R3. Also, it is worth mentioning that the here proposed approach could be integrated in a graph representation similarly to our previous works (ref.^{10,11}), so that the dynamical regime of the system would be tuned also depending on the spatial position of multiple input signals, even if this is out of the scope of this work.

Changes in the manuscript: new results on the implementation of computing have been included in the revised version of the manuscript (refer to response R3). In addition, it has been clarified in the revised manuscript (page 27-28) that the here proposed approach could be integrated in a graph representation to model also the emergent behavior of multielectrode networks.

C9. Figs. 3 and 4 could be combined into one figure to more clearly compare the modelling performance.

R9. We thank the reviewer for pointing out this aspect. To provide a clearer comparison of modeling results with experimental data we have included a new Supplementary Figure in the revised manuscript allowing the direct comparison between experimental and modeling results on multiple timescales. Since combining Figures 3 and 4 into one figure would result in a figure with a large number of panels, in our opinion adding a Supplementary Figure clarifying this aspect would increase the readability of results.

Changes in the manuscript: a new Supplementary figure has been added in the revised manuscript (Supplementary Figure S4). A sentence pointing out that a direct comparison of experimental and modeling time traces on multiple timescales is reported as a supplementary information has been added in the main manuscript (page 15).

C10. As a modelling work, it is recommended to provide the simulation model that simultaneously describe deterministic and stochastic processes to facilitate the followers to understand the details.

R10. The simulation model exploited in this work have been made publicly available on GitHub at the link:

https://github.com/MilanoGianluca/Self-organizing_neuromorphic_networks_as_stochastic_dynamical_systems

In addition, the whole dataset of the work will be made freely available on Zenodo, and crosslink between manuscript and dataset will be provided during the publication process.

Changes in the manuscript: A code availability statement has been added, with the link to the GitHub repository.

Reviewer #3 (Remarks to the Author):

In this manuscript Milano et al describe the dynamical properties of nanowire networks within a framework of emergent behaviour and neuromorphic computing, and attempt a holistic description of deterministic and stochastic effects based on Ornstein-Uhlenbeck (OU) processes. This is an interesting topic and is very relevant to a growing literature on nanoscale networks. Unfortunately I have a number of concerns.

We thank the reviewer for finding the topic interesting and relevant, and for the constructive comments that gave us the possibility to improve our work.

C11. The aim is to “demonstrate that nanowire-based neuromorphic networks are stochastic dynamical systems where the signals flow relies on the *intertwined* action of deterministic and random factors” and there is an emphasis throughout the manuscript on the importance of the internal dynamics of the system and of emergent behaviour. My understanding is that the intention is to demonstrate the rich dynamics, which might result from the coupling (intertwining) of the three terms in eq 1. Hence my main concern is that I do not see clear evidence in the experimental data for internal dynamics of the system, emergent behaviour, or for coupling.

R11. What the model we are proposing represents is the superposition of three simultaneous processes occurring at different time scales. Let’s leave aside the jumps which are a particular case of fluctuations (large fluctuations following a Poisson process) and let’s concentrate on the deterministic term and the low-level fluctuations. Experimental data reveals that in the long term, the system’s conductance stabilizes around a value that follows a sigmoidal trajectory as a function of the applied bias. The fact that the conductance stabilizes gives a first hint indicating that irrespective of the noise amplitude (which is assumed constant for simplicity) and initial condition, the average conductance converges towards a given value (so-called attractor). This is precisely the mean-reverting property of the OU process (see Figs. 1 and 2). The sign of the time derivative of the analyzed variable (conductance) is positive or negative depending on the magnitude of the difference between the variable itself and the long-term expectation. Now we add noise. If we solely considered noise without a mean-reverting term, then the stochastic process diverges (since we are defining the process in terms of a differential equation, i.e. as a dynamical system). Which noise and how should it be included? Well, this is a matter of discussion. We have selected additive Wiener noise because this is the standard approach and because we know it provides a stable solution, but more complex options are possible indeed (Levy flights?). What the OU process enables is the generation of a Gaussian process with bounded variance and autocorrelation (see Fig. 3 and new Supplementary Figure S2) which ultimately leads to a stationary probability distribution. The sigmoidal trajectory exhibited by the attractor’s states (model and experiments) (Fig. 2b) is a consequence of the physical aspect of the problem (diffusion and volatility) and the assumed mathematical approach. Of course, this is certainly an approximation of a more complex behavior whose ultimate origin is inaccessible since it arises from the interaction at the nanoscale of a huge number of junctions. The proposed approach is the simplest Markov-Gaussian process one can postulate, where the emergent behavior and coupling are inherent to the assumed description. Please, refer to response R12 for further details on the coupling of deterministic and stochastic dynamics.

Changes in the manuscript: we have clarified through the text that our approach represents the simplest Markov-Gaussian process that can be postulated, and that emergent behavior and coupling are inherent to the assumed description (page 6-7 of the revised manuscript). Additional results to clarify the intertwined action of deterministic and random effects are reported in the new Supplementary Figure S2.

C12. Perhaps even more fundamental to the manuscript is whether OU processes, which are typically walk type processes, can lead to the kind of dynamics and emergent behaviour that might be useful for neuromorphic computing. It seems to me essential to discuss this point in the manuscript. Descriptions of stochastic dynamical systems in the literature make it clear that OU systems can be interesting as noise can stabilize unstable equilibria, shift bifurcations, and can lead to transitions between coexisting deterministic stable states. But such effects are not clearly demonstrated in the manuscript at the moment.

R12. We agree with the reviewer that the connection between OU dynamics and computing was not clarified in the original manuscript. We have expanded our work to connect the proposed description of the network as a stochastic dynamical system and its information processing capabilities in the framework of reservoir computing (evaluating computing performances on standard benchmark computing tasks), showing that the OU dynamics well describing the emergent behavior of the NW network can be exploited for neuromorphic type of computing. Due to the parabolic nature of the potential, we cannot expect bifurcations or transition between stable states. These effects are not inherent to a mean-reverting type OU process. In this context, it is worth noticing that the scope of the work was not to demonstrate shift bifurcations nor the transitions between coexisting states, but rather to develop for the first time a novel modeling framework able to describe, in one compact equation, the deterministic and stochastic evolution of the memory state of the network when subjected to arbitrary input signals, a fundamental aspect for testing the implementation of unconventional computing paradigms.

Changes in the manuscript: new results showing how the emergent behavior might be useful for computing are reported Figure 8 and 9 as well as in new Supplementary Figures S13 and S14 and discussed in detail at pages 23-26 of the revised manuscript. All details on the computing implementation are reported in new methods sections and in new Supplementary Figure S12. All these aspects were clarified through the whole manuscript, including abstract, introduction, discussion and conclusions.

C13. Throughout the paper it appears that the deterministic effects are analysed quite separately from the noise and jumps. For sure there are transient effects – when the input is changed, the conductance slowly evolves towards a steady state value – but I do not see evidence for coupling of the conductance changes to the noise or jumps. In fact Fig 1 seems to make it clear that the noise / jumps are *independent* of the conductance (and in Fig 5 the noise / jumps appear to simply be superimposed onto the deterministic behaviour). If they are dependent this needs to be clearly shown. Can the authors identify features of the experimental data which could not be explained by a simple model of a voltage controlled resistor (with sigmoidal dependence on V) together with *independent* noise / jumps?

R12. Thank you for this very interesting point. According to our understanding the question is why we don't simply consider a deterministic model for the conductance and we add a noise term. By considering the autocorrelation plot of experimental conductance time trace of a network in the

stationary state, results show that sampled data are not independent (refer to the new Supplementary Figure S2). This means that the conductance time trace cannot be modeled simply by superimposing noise to a deterministic trajectory. Indeed, in a stationary state no autocorrelation is expected if deterministic and stochastic dynamics are not coupled (i.e., simply adding noise to a deterministic trajectory in a stationary state does not result in autocorrelation). For this reason, we considered an Ito-type equation (not a deterministic solution plus noise). We adopted the standard approach of representing the system evolution in terms of a first order differential equation, i.e. a dynamical system. This is more convenient when we have changing constraints (voltage) and when the new initial condition is automatically derived from the last calculation). In this framework, we have assumed as a first approximation (and in qualitative accordance with respect to experimental results) a constant noise with respect to the memory state variable g (i.e., σ is not dependent on g). Of course, this can be more elaborated than assumed, but this is the first attempt to describe such a complex system.

Changes in the manuscript: we clarified through new experimental data evidence for coupling of conductance changes to noise/jumps by adding the autocorrelation plot of the conductance time trace (new Supplementary Figure S2), adding a sentence recalling this point at page 6 of the revised manuscript. We also clarified through the text that we have assumed as a first approximation a constant noise with respect to the memory state variable g (page 6).

C14. In what way does conductance g represent internal *memory* of the system? The conductance of the system is controlled by the applied voltage (Fig 2b showing sigmoidal behaviour). On p19 it is stated that “enhanced dynamics of the internal memory state (i.e., higher dynamical range of g) can be observed...”, which relates to the data in Fig 7. From the data presented it appears that the higher dynamical range results simply from the differences in the gradient of the sigmoidal response to the applied triangle wave i.e. the deterministic response has a steep gradient at 3.6V and almost zero gradient at 5V. If these oscillations were related to the internal dynamics of the network, would they not have similar timescales to the transients rather than the period of the driving waveform?

R14. The (normalized) conductance g represents the internal *memory* of the system since its value depends on the history of applied electrical stimulation. In these terms, the conductance of the system depends not only on the applied voltage but also on the recent history of electrical stimulation because of the short-term memory capabilities of the system. This short-term memory capabilities, that can be exploited for processing information over time, arises from volatile switching mechanism (refer to our previous works for a detailed description of physico-chemical mechanisms underlying short-term memory effects in NW networks^{7,12}). In this context, it is important to remark that, besides the attractor state, also internal dynamics of the network rely on the bias voltage since different voltages lead to different reversion speeds, as detailed in a new supplementary image (Supplementary Figure S7). In this context, higher reversion speeds results in outputs that respond faster to the input, while a lower reversion speed results in an output that respond slower to the input. Moreover, the response speed relies on the input signal frequency and amplitude of the signal as well. These factors should be analyzed depending on the specific input signal.

Changes in the manuscript:

- It was clarified through the manuscript that g represent the internal *memory* of the system that evolves over time depending on the history of applied electrical stimulation (page 6 of the revised manuscript).
- It was clarified that not only the attractor state but also network dynamics are affected by the applied voltage, since different voltages lead to different reversion speeds (page 20). For this purpose, a new supplementary figure (Supplementary Figure S7) clearly showing the dependence of the reversion speed on the bias voltage has been included in the revised version of the manuscript.

C15. A related point is the use of the word “attractor”. From wikipedia: “In the mathematical field of dynamical systems, an attractor is a set of states toward which a system tends to evolve, *for a wide variety of starting conditions of the system.*” Rather than demonstrating attractor states, the presented experimental data seems to show that each applied input voltage (i.e. the starting conditions) tunes the system to a *different* state. It would be great if the authors could clearly show that the system exhibits attractor states.

R15. We agree with the reviewer that this aspect was not correctly detailed through the manuscript. For this purpose, we reported new data as supporting information showing that the system exhibits attractor states as a function of the applied voltage. Of course, we cannot get rid of certain variability which a more elaborated version of the proposed model should account for.

Changes in the manuscript: we reported new experimental data showing that the system exhibits attractor states in a new supplementary figure (Supplementary Figure S1). This aspect was clarified in the revised manuscript (page 5-6).

C16. Similar points can be made in relation to the mathematical argument presented. It is stated that “when jumps are included, the stochastic differential equation needs to be solved numerically through the Euler–Maruyama method (details in Methods).” This would indeed be the case if complex dynamics were observed, but to the best of my understanding in this paper the deterministic part of the equation is simply integrated to give the average rate of change of the conductance (θ) and this is done completely separately from the analysis of the noise (eq 7). Perhaps this point is intended to be clarified in the methods but I found the text on p26-27 very unclear. This point is vital for the paper so it is really important that this argument is clarified and a careful summary of the detailed argument was added to the main text. It would help a lot if figures were added illustrating the process described on p26 using real experimental data. Similarly I found the text on p27 describing the construction of the energy U to be unclear with some specific points of confusion (how can a derivative with respect to conductance be equal to a derivative with respect to conductance (the dimensions are surely not correct) and where does this equation come from?).

R16. Concerning the jumps, we adopted the Euler-Maruyama method to approximate the numerical solution of the stochastic differential equation. In this case, jumps appear as an additive term to this solution. Eq.(1) is just a convenient representation, but the actual meaning is that steps occur during the time evolution of the variable. Jumps in the conductance time trace are generated by jumps in the differential equation reported in eq. (1). In our case, jumps in the conductance time trace should not alter the process variance, only the long-term expectation (at the end compensated by negative steps). This is consistent with the initial assumption of a constant noise term independent of the magnitude of the signal. Of course, more complex dynamics can be considered, but results show that the proposed model is able to describe main features of deterministic and stochastic network dynamics.

In an OU process, to determine the deterministic trajectory, we eliminate the noise term in the Ito-type equation and integrate it. This is the standard approach (refer for example to ref.¹³), where noise is usually characterized in the stable region of the trajectory. The disentanglement between noise and jumps is also carried out in the stable region of the curve for simplicity. This is what we have shown and discussed along this work. Of course, we can simply add noise to the deterministic trajectory, but this does not yield autocorrelation as exhibited by our system (see detailed discussion on this aspect in response R12, with new experimental results in Supplementary S2)

Concerning the construction of the energy U , this is a standard approach when analyzing the dynamical aspect of a system. Note that g is dimensionless quantity so that U refers to a mathematical potential landscape whose minimum indicates the equilibrium state. In this framework, the minimum of U in terms of g (null slope) corresponds to a stationary value of g . This potential is relevant when solving the probability distribution of the stationary state by means of the Fokker-Planck equation.

Changes in the manuscript: all these aspects were clarified through the manuscript, adding also new experimental results to clarify the coupling of deterministic and stochastic dynamics as new Supplementary Figure S2). The process described has been further clarified using experimental data reported in Supplementary Figure S2.

C17. I found it difficult to understand the nature of the noise and jumps as the data is not presented on scales that allow examination of the data. The only figure that shows the noise and jumps clearly is Fig 6c, and there the noise appears to be almost sinusoidal, which seems to be quite different to that in Figs 1 and 5a, although the scale in earlier figures makes it difficult to be sure. Even in Fig 6c it would help the reader a lot if there were zoomed in plots that allow the response to a jump to be visualized.

R17. We thank the reviewer for pointing out this aspect. To facilitate the examination of data for the reader, we have included a new supplementary information in the revised manuscript to directly compare experimental and modeling results on multiple timescales (new Supplementary Figure S3). The normalized conductance time trace reported in Figure 6c was improved to enhance data visualization. Also, a zoomed view of Figure 6c to better visualize the jump response is provided as new supplementary information (Supplementary Figure S7). In these figures it is possible to observe that the signal is not a sinusoidal-like noise but is the result of the mean-reverting process, i.e., what we are ultimately proposing.

Changes in the manuscript: a new Supplementary figure has been added in the revised manuscript (Supplementary Figure S3) and a sentence pointing out that a direct comparison of experimental and modeling time traces on multiple timescales is reported as a supplementary information has been added in the main manuscript (page 15). The enlarger view of the conductance time trace reported in Figure 6c is reported in the new Supplementary Figure S7.

C18. Throughout pages 17-19 I was unable to tell which data / comments related to experiments and which to simulations. This needs to be clarified in all the captions as well as the text. I am guessing, but I think perhaps Fig 6c may be from the simulations, in which case comparison with similar zoomed in plots of the experimental data are essential to understand how reliable the characterization of noise and jumps are, and to allow an understanding of whether the simulation and experiments are similar or not. I suggest that the authors show segments of data from Figs 3a, d and g on much shorter timescales. I was surprised to find a comment in the methods section that Fig 7 is from simulations rather than experiments and I wonder why comparable experimental data are not presented. A clear comparison between experimental and simulated data would help the reader enormously.

R18. According to reviewer suggestions, we have clarified through the whole manuscript which data/comments are related to experiments and which to simulations. As correctly pointed out by the reviewer, Figure 6c is from modeling results since the aim of this figure is to provide a description of the potential landscape corresponding to the assumed model. As requested by the reviewer, in order to allow a better comparison on multiple and shorter time scales of noise and jumps in experimental and modeling time, a new supplementary information figure has been added (refer to response R17).

Results reported in Figure 7 are from simulations, since the scope of this figure was to show that the model allows to disentangle the effect of deterministic and stochastic dynamics on the evolution of the internal memory state of the network.

Changes in the manuscript: it was clarified through the whole manuscript which data/comments are related to experiments and which to simulations and we have provided segments on much shorter timescales of Figure 3 and 4 as a new supplementary figure (Supplementary Figure S4) to provide a direct comparison of noise and jumps in experimental and modeling results. A sentence recalling this aspect was added in the revised manuscript (page 15).

Some more technical points:

C19. The data presented in fig 1d and e does not clearly show that an equilibrium is reached and so data should be shown for a longer time period.

R19. The reviewer is right, data reported in Figure 1d and e does not clearly show that the equilibrium is reached and was somehow misleading for the reader. The aim of this introductory figure is to show system dynamics characterized by combined deterministic and stochastic effects. For this purpose, we have revised the figure by removing reference to the attractor state and the text has been revised accordingly. Instead, the complete analysis of attractor states reported in Figure 2 was performed after long-term stabilization ensuring that the system have reached the equilibrium state (refer to the inset of Figure 2b).

Changes in the manuscript: Panels d and e of figure 1 were revised by removing graphical reference to the attractor state, the figure caption was revised accordingly. The manuscript has been revised for avoiding misunderstanding (page 5-6).

C20. How can the noise in Fig 1f be clearly distinguished from the jumps? Isn't it possible that the noise is just smaller jumps?

R20. The disentanglement of noise and jump events have been performed through a thresholding algorithm that maximizes the p -value of the Gaussian distribution of the noise component of the signal. In particular, the disentanglement has been performed by determining the interval underlying the most gaussian data region in a Kolmogorov-Smirnov (KS) sense. In details the p -values of progressively larger data intervals (centered in 0) have been evaluated by using the KS test. The extreme ensuring maximum p -value, i.e. most probable gaussian distribution, has been chosen as the threshold between noise and jump events. In this context, it is worth noticing that in our modeling approach we are not only considering the magnitude of fluctuations but also their temporal aspects, where jumps are considered “rare” events. Refer to response R1 and R2 to reviewer #1 for a detailed discussion concerning the origin of noise and jump events.

Changes in the manuscript: The aspects, also discussed in response R1 and R2, were clarified in the revised manuscript, at page 6, clarifying the difference between noise and jumps, with reference to previous works¹.

C21. It is stated that Fig 2b shows a long tailed distribution and (in the methods) that logarithmic bins are used. Neither of these appears to be correct and furthermore the logarithmic y axis in b makes it

impossible to compare with Fig 2e. Fig 2h does not look like an exponential – why is the data not shown on log linear scales?

R21. In his/her comment, we guess that the reviewer is referring to figure 3 and not figure 2. Here, we agree that the text (manuscript and methods) was somehow misleading. In that case, Figure 3b shows the histogram of dG/dt with logarithmic y axis with linear binning, not with logarithmic bins. The aim of the logarithmic y axis was to clearly show the long-tailed distribution, while a direct and quantitative comparison of distribution reported in Figure 3b and 3 can be seen from the direct comparison of their corresponding quantile-quantile plots reported in Figure 3c and f, respectively. Here, the quantile-quantile plot corresponding to the histogram reported in Figure 3b (Figure 3c) clearly shows that compared to the normal distribution there is much more data located at the extremes of the distribution and less data in the center of the distribution (heavy tailed qq-plot). Instead, the quantile-quantile plot relative to the noise component of dG/dt reported in Figure 3e (Figure 3f) is in good agreement with the theoretical Gaussian distribution (i.e., the linearity suggests that data are normally distributed). Similar considerations holds for modeling results reported in Figure 4b,c,e and f. Notably, a good agreement with experimental distributions and modeling results can be observed.

According to the reviewer suggestion, we report data of Figure 3h (and Figure 4h) in log linear scales. As can be observed in Figure 3h, the behavior can be described through an exponential dependence in first approximation, where deviations from the exponential behavior for high IEI can be ascribed to small data representation in this regime due to the lower occurrence probability of these events. Also in this case, a good agreement of experimental and modeling results can be observed.

Changes to the manuscript: we have clarified in the revised manuscript that the quantile-quantile plot allows the direct observation of the heavy tailed distribution (page 12). Also, we have replotted Figure 3h (and also 4h) in log-linear scales in the revised manuscript according to the reviewer request.

C22. The equation for Poisson processes on p14 surely cannot really be $N(t)=t$? What is the meaning of $\lambda=1$?

R22. In a nonhomogeneous Poisson process the number of events at time t is often represented by the equation $N(t) = \lambda t^\gamma$. According to experimental results, $N(t)$ is shown to linearly increase over time (refer to experimental data in Figure 3c) as expected for a homogeneous Poisson process so that $\gamma = 1$. Instead, λ represents the event rate of the Poisson process, i.e., the slope of the $N(t)$ vs t relationship that according to experimental results was found to be $\lambda \sim 0.082$ events/s (not 1).

Changes in the manuscript: it was further clarified through the text that $\lambda \sim 0.082$ events/s according to experimental results (page 15 of the manuscript).

C23. Data should be presented to allow the MLE fitting to be understood.

R23. Assuming that the reviewer is referring to data reported in figure 3g and 4g, we have reported an enlarged view of data of these figures as a supplementary image allowing easier examination of data interpolation with a straight line. This is a least-squares fitting problem.

Changes in the manuscript: a new supplementary image (Supplementary Figure S4) has been included in the revised version of the manuscript for clarifying this aspect to the reader. A sentence recalling this new supplementary image has been added in the revised manuscript (page 15).

Finally, while the manuscript is generally clearly written, if the authors intend to revise their manuscript they should also revise some of the choices of wording. Examples include:

C23. Is software really aim of the neuromorphic computing community?

R23. We agree with the reviewer that software is not the original core aim of the neuromorphic computing community, so we revised the text accordingly. It is important to remark also that the neuromorphic computing community should also be aware of software because this establishes benchmarks.

Changes in the manuscript: we have modified the first sentence of the abstract from “Neuromorphic computing aims to develop software and hardware platforms ...” to “Neuromorphic computing aims to develop hardware platforms ...”.

C24. Terms like “self-organized synergy” and “stochastic emerging behavior” seem meaningless and should be removed.

R24. To avoid misunderstanding, we have modified the text according to reviewer suggestions.

Changes in the manuscript: the term “self-organized synergy” has been modified to “self-organized behavior” (page 3 of the revised manuscript), the term “stochastic emerging behavior” has been modified to “stochastic behavior” (page 4 of the revised manuscript).

C25. “Reversion speed” is not explained – it seems to be simply a rate of change.

R25. We agree with the reviewer that this concept was not clarified through the text. In the context of Ornstein–Uhlenbeck process, the reversion speed (θ) represents the speed of reversion of the process to its long-term mean, i.e., the rate at which the internal memory state of the system reverts towards the attractor state. An OU process is also called mean-reverting.

Changes in the manuscript: it was clarified that the reversion speed represents the rate by which the internal memory state of the system reverts towards the attractor state at page 6 of the revised manuscript.

C26. Why is the network conductance “effective”?

R26. We agree with the reviewer that this aspect was not properly clarified through the text. In our work, the conductance between two areas of the network is considered. This can be referred as “effective” conductance in the sense that it represents the conductance “seen” by the two areas where a voltage difference is applied. In the framework of circuit networks, it is usual to refer to “effective conductance” when considering the resulting conductance in between two network nodes connected by multiple series/parallel conductive pathways. In our case, the resulting effective conductance in between stimulated areas (and its evolution over time) relies on a high number of series and parallel current pathways composed of multiple memristive junctions, as investigated in detail through graph theory in our previous work.¹⁰

Changes in the manuscript: the above-mentioned discussion was added at page 5 of the manuscript, clarifying that the network (effective) conductance is the result of multiple series and parallel pathways. To avoid confusion for the reader we substituted “effective conductance” with “conductance” through the whole text.

References

1. Diaz-Alvarez, A. *et al.* Emergent dynamics of neuromorphic nanowire networks. *Sci Rep* **9**, 14920 (2019).
2. Zhu, R. *et al.* Information dynamics in neuromorphic nanowire networks. *Sci Rep* **11**, 13047 (2021).
3. Milano, G., Cultrera, A., Boarino, L., Callegaro, L. & Ricciardi, C. Tomography of memory engrams in self-organizing nanowire connectomes. *Nat Commun* **14**, 5723 (2023).
4. Manning, H. G. *et al.* Emergence of winner-takes-all connectivity paths in random nanowire networks. *Nat Commun* **9**, 3219 (2018).
5. Atiya, A. F. & Parlos, A. G. New results on recurrent network training: unifying the algorithms and accelerating convergence. *IEEE Trans Neural Netw* **11**, 697–709 (2000).
6. Demis, E. C. *et al.* Nanoarchitectonic atomic switch networks for unconventional computing. *Jpn J Appl Phys* **55**, 1102B2 (2016).
7. Milano, G. *et al.* In materia reservoir computing with a fully memristive architecture based on self-organizing nanowire networks. *Nat Mater* **21**, 195–202 (2022).
8. Milano, G., Montano, K. & Ricciardi, C. In materia implementation strategies of physical reservoir computing with memristive nanonetworks. *J Phys D Appl Phys* **56**, 084005 (2023).
9. Appeltant, L. *et al.* Information processing using a single dynamical node as complex system. *Nat Commun* **2**, 468 (2011).
10. Milano, G., Miranda, E. & Ricciardi, C. Connectome of memristive nanowire networks through graph theory. *Neural Networks* **150**, 137–148 (2022).
11. Montano, K., Milano, G. & Ricciardi, C. Grid-graph modeling of emergent neuromorphic dynamics and heterosynaptic plasticity in memristive nanonetworks. *Neuromorphic Computing and Engineering* 0–22 (2022) doi:10.1088/2634-4386/ac4d86.
12. Milano, G. *et al.* Brain-Inspired Structural Plasticity through Reweighting and Rewiring in Multi-Terminal Self-Organizing Memristive Nanowire Networks. *Advanced Intelligent Systems* **2**, 2000096 (2020).
13. Ait-Sahalia, Y. Disentangling diffusion from jumps. *J financ econ* **74**, 487–528 (2004).

Point-by-point response

We thank the reviewers for the general positive assessment of our work and for recognizing significant improvements in the revised version of the manuscript. Following reviewers' #2 and #3 comments, we have largely revisited the manuscript and related supplementary information to further improve their contents (changes in this revision phase are highlighted in red) and to shed more light on a number of aspects. We believe that the current manuscript, which includes additional experiments and analyses, is in line now with the high publication standards required by *Nature Communications*.

Reviewer #1

The paper reporting the stochastic and deterministic behaviour of the time-dependent conductance of the voltage-stimulated Ag nanowire network with an interpretation using the Ornstein-Uhlenbeck (OU) process will certainly provide insights towards better understanding and achieving controlled exploitation of complex networks. Due to the complexity of the material system, it's usually difficult to understand the whole behaviour of such a system (complex network) without computation under certain conditions, but the authors have shown the possibility to formulate the behaviour with a minimal number of assumptions.

Although there are still open questions (probably difficult to solve at the moment), I believe that the paper is worthwhile to be published in its present form, which would result in stimulating related research towards future computation. And I expect that the work would be cited as one of the milestones in the course of developing neuromorphic computing that does not consume energy as has been developed so far within the framework of current computing schemes based on von Neumann architecture.

Reviewer #1 (Remarks on code availability):

I do not have sufficient background to review the code itself. Sorry.

We thank the reviewer for the positive assessment and for finding our revised work worthwhile of publication in *Nature Communication*.

Reviewer #2

Substantial improvements have been made in the revised manuscript, especially the additional simulation for demonstrating the performance in reservoir computing. There is one remaining concern before it can be considered for publication in Nature communications.

C1. In the authors' response R3(3), the authors mentioned "3) We quantitatively connected deterministic and stochastic dynamics with computing performances". By learning Fig. 7-9 and the relevant text, I feel like the simple fact is that the deterministic evolution of the memory state provides much less noisy dynamics than the stochastic evolution (Fig. 7c-f), resulting in a lower NMSE (Fig. 8c and 9c). Meanwhile the rest figures regarding bias, theta and N, are more like an optimization strategy, as reservoir computing is known to be highly sensitive to those parameters. The way that the authors connect the det&sto with computing performances is simply tuning the relevant parameters. It can only be concluded from these results that a less noisy system yields a lower NMSE. The authors should give more in-depth insights on the effect of det&sto and the underlying mechanism.

R1. We thank the reviewer for pointing out this aspect of the proposed approach. Following the reviewer's suggestion, we provide in the revised manuscript a more in-depth discussion about the impact of deterministic and stochastic effects on computing performance, as detailed next.

Firstly, we added more insights into how the choice of N and Θ parameters for the minimization of NMSE relies on the operating condition in terms of polarization voltages by considering deterministic dynamics. As an example, Figure R1 reports new results showing the effects of N and Θ on the NARMA task performance depending on the bias voltage. When operating the network away from state $\tilde{g} \sim 0.5$, besides a general increase of NMSE, a substantial decrease in performance can be observed when considering high N or Θ values, i.e., in cases where network dynamics are expected to be largely affected by a reduction of the fading memory properties.

Secondly, we added new results demonstrating, differently from the deterministic case, that stochastic effects lead to computing performances that are highly dependent on the masking scheme adopted for time-multiplexing. Figure R2 and R3 report results in that direction showing the effect of alternative masking schemes on computing performances simulated by considering the deterministic and stochastic model, for the NARMA-2 and sine-to-cosine waveform NLT transformation tasks, respectively. As can be observed, when stochastic effects are taken into account, performance is highly degraded for some specific choices of N and Θ values. While the choice of the mask is not substantially affecting the system in the deterministic case, results show that the specific N and Θ values with degraded performances rely on the specific masking scheme adopted for time multiplexing. This is because each specific masking scheme combined with the input signal generates different network dynamics, driving the network in a dynamic regime that endows information processing capabilities that can be less or more resilient to stochastic effects. These results show that stochastic effects should be properly considered for the optimization of time-multiplexed reservoir computing scheme for the proper selection not only of $[N, \Theta]$ parameters but also for the design of the masking scheme for the signal input. As a result, the masking scheme can be optimized to reduce degradation of computing performances for a given set of $[N, \Theta]$ through, for example, supervised learning algorithms. In this framework, it is important to point out that stochastic effects do not hinder the possibility of operating the system with a low number of virtual nodes N , as required for reducing the hardware complexity of the system. Indeed, a lower N number allows the system to operate with a reduced number of weights to be trained (and a lower amount of information to be temporally stored before being analyzed by the readout).

Changes in the manuscript: According to the above discussion, we have strongly revised the section discussing the relationship between attractor states and computational capabilities, as well as the discussion paragraph, as detailed below:

- We have included in the revised manuscript a new section providing more insights on how the choice of N and Θ parameters for the minimization of NMSE relies on the operating condition and peculiar network dynamics (page 24). Here reported Figure R1 has been added as a new supplementary figure (Supplementary Figure S14).
- We have included in the revised manuscript a new section providing more insights showing that, differently from the deterministic case, the effect of stochasticity leads to computing performances that are highly dependent on the masking scheme adopted for time-multiplexing reservoir computing (pages 24 and 25). Here reported figures R2 and R3 have been added as new supplementary figures (Supplementary Figure S15 and S18).
- We provide more insights concerning deterministic and stochastic dynamics on computing capabilities in the revised discussion paragraph, page 29 and 30 of the revised manuscript.

Figure R1 | Effect of N and Θ on the NARMA task performance depending on the polarization voltage. Colormaps showing NARMA-2 task performances simulated with the deterministic model as a function of polarization voltages of **a.** 2.4 V, **b.** 2.9 V, **c.** 3.6 V ($\tilde{g} \sim 0.5$), **d.** 4.3 V, and **e.** 4.8 V, by stimulating the network with an input amplitude of 50 mV. A substantial decrease in performance can be observed by considering sets of parameters with high N or high Θ values when operating the network away from $\tilde{g} \sim 0.5$, since in these cases network dynamics are more affected by a reduction of fading memory properties.

Figure R2 | Effect of the masking scheme during time multiplexing on the NARMA-2 task performance. a-d. Different masking schemes and corresponding colormaps showing task performances as a function of N and θ parameters simulated by exploiting the e-h. deterministic and i-l stochastic model. Results have been obtained with a polarization voltage of 3.6, operating the network at $\tilde{g} \sim 0.5$. While the masking scheme does not substantially affect performance in terms of NMSE in the deterministic case, mask-dependent degradation of performances can be observed for some specific sets of $[N, \theta]$ parameters. This indicates that the peculiar dynamics of the network induced by the masking scheme combined with the signal input can lead to computing performances of the system that can be less or more affected by stochastic effects.

Figure R3 | Effect of the masking scheme during time multiplexing on the sine to cosine waveform NLT task performance. **a-d.** Different masking schemes and corresponding colormaps showing task performances as a function of N and θ parameters simulated by exploiting the **e-h.** deterministic and **i-l** stochastic model. Results have been obtained with a polarization voltage of 3.6, operating the network at $\tilde{g} \sim 0.5$. While the masking scheme does not substantially affect performance in terms of NMSE in the deterministic case, mask-dependent degradation of performances can be observed for some specific sets of $[N, \theta]$ parameters. This indicates that the peculiar dynamics of the network induced by the masking scheme combined with the signal input can lead to computing performances of the system that can be less or more affected by stochastic effects.

Reviewer #3

The authors have made some significant revisions to the manuscript, including adding a new section on computation, but I was unable to see changes that address the majority of my original comments.

C2. As with the original manuscript my main concerns relate to the central claim in the abstract “we demonstrate that *nanowire-based neuromorphic networks* are stochastic dynamical systems where the signals flow relies on the *intertwined* action of deterministic and random factors” and comments in the conclusions about the importance of emergent behavior. In my view, since the nanowire networks are experimental devices, to justify their claims the authors need to provide *experimental* data that support their claims, and specifically data that support the claims that the deterministic and random factors are intertwined, and lead to emergent behavior. Much of the manuscript relates instead the authors’ model and the connection to the experimental system is not nearly clear enough.

R2. We thank the reviewer for pointing out this particular aspect of the system’s behavior again, which apparently was not sufficiently explained. We agree with the reviewer that the text related to the intertwined action of deterministic and random factors as well as the use in the manuscript of the concept “emergent behavior” can be somehow misleading for the reader. In what follows, these issues are further discussed and according to our viewpoint clarified. These points have been also accommodated in the main text.

Connection of model and experimental system

Concerning the deterministic dynamics, we show that experimental data exhibits the mean-reverting property typical of an OU process. In this connection, we show that the deterministic dynamics described by our model correctly addresses the experimentally observed dependence of the stationary conductance state (\tilde{G}) as a function of the applied voltage bias (Figure 2b). In addition, we show that the model can describe the experimental time traces (Figure 2a and 2c).

Concerning the stochastic dynamics, we show that experimental data can be well described by a mean-reverting process with noise and jumps where deterministic and stochastic dynamics are inherently intertwined. We would like to emphasize that, even if noise is analyzed in the stationary state to avoid the initial transient, we do not describe noise and jumps independently from deterministic dynamics, since the stochastic dynamics expressed by eq. (1) is inherently coupled to the deterministic mean-reverting process. In fact, we show that the main features of the experimental characteristics, including experimental dG/dt distribution of noise, number of jump events $N(t)$ as a function of time, probability distribution of inter-event intervals (IEIs) as well as probability distribution for the jump amplitude Γ can be well described by the proposed model endowing both deterministic and stochastic features (Figure 3 and 4).

In the revised manuscript, we strengthened the connection between the experimental system and the stochastic model by further analyzing the autocorrelation function of experimental data. An exponential dependence as a function of the lag number was obtained, which is precisely the characteristic of an OU process (see response R5). In addition, we provide additional results showing a direct comparison between experimental and modeled conductance time traces, providing also detailed experimental and simulation examples of the mean-reversion effect after a jump (see response R11).

While our approach certainly has some important limitations (for example it cannot describe long-term changes in the network conductance such as those experimentally observed under specific conditions,¹ as now detailed in the revised manuscript at page 28), the proposed model is based on a

combination of physical foundations and mathematical properties that allows describing the behavior of the system in terms of a minimal set of assumptions (as pointed out also by reviewer #1). To the best of our knowledge, this represents the first attempt to describe with a single equation both deterministic and stochastic dynamics of a self-assembled memristive system.

Emergent behavior

In self-organizing memristive systems, the concept of “emergent behavior” typically describes collective phenomena arising from local nanoscale effects caused by the interaction of a multitude of nano-objects (refer to ref.²). This concept has been associated with the electrical response of NW network under electrical stimulation, where the network conductance relies on the memristive interaction of a multitude of NWs and NW junctions (refer for example to ref.³). In this sense, the experimentally observed conductance time traces of the NW network are the result of the emerging behavior of the system, as experimentally investigated in ref. ⁴. In these terms, the proposed approach is exploited to model the emergent behavior of the network in terms of conductance dynamics under electrical stimulation.

The concepts have been clarified in the revised manuscript, as detailed in responses to comments below.

Below I clarify why I do not think my original comments have been addressed. Note that while my comments mostly refer to the rebuttal I want to emphasize that these points really *have* to be clear *in the manuscript*.

C3. C11: In R11 the authors state “that emergent behavior and coupling are inherent to the *assumed description*”. I have no doubt that *the model* can exhibit emergent behavior, but I do not see evidence for this in the experiments. I think it is essential that the authors clarify what evidence there is for these effects *in the experimental data*.

R3. As discussed in response to C2, conductance time traces reported in our work results from the emergent behavior of the NW network in the sense that conductance dynamics (both deterministic and stochastic) rely on collective phenomena involving a multitude of switching events in NW and NW junctions. This concept of emergent dynamics in NW networks was discussed in previous works not only from our group,⁴⁻⁶ but also from others ^{2,3}.

Changes in the manuscript:

- We have clarified the concept of “emergent behavior” (as detailed response R2) in the introduction of the revised manuscript, while relevant references supporting the discussed concept have been added (page 3 of the revised manuscript).
- We have further clarified that the experimental conductance time traces are the result of the emerging behavior of the system related to collective phenomena (page 5 of the revised manuscript).

C4. I was confused by R12 (“the scope of the work was not to demonstrate shift bifurcations nor the transitions between coexisting states”) which seems to suggest that even the model is in a regime where emergent behavior is not observed. Is that correct? All of this needs to be clearly discussed in the manuscript.

R4. According to the concept of emergent behavior explained in responses R2 and R3, we developed a novel modeling framework able to describe and simulate the system’s response that is here shown

to be consistent with an OU process. In other words, we show that the OU process can describe emergent dynamics of the system understood as the result of uncountable interactions hidden to the external observer. This emergent behavior is not expected to exhibit shift bifurcations (in the sense of chaotic behavior). On the other hand, transitions among coexisting states are observed in certain conditions but were not modelled. This is because to implement jumps between coexisting states we would have to introduce new parameters able to account for modifications of the attractor states. This only adds more complexity and is irrelevant to what we want to highlight in the current phase of our work.

Changes in the manuscript: we have clarified in the revised manuscript that our modeling approach, characterized by a parabolic potential, does not endow bifurcations or transitions among coexisting deterministic stable states (page 18 of the revised manuscript). We clarified in the revised discussion paragraph that our modeling approach can be further expanded by introducing transitions among coexisting states and long-lasting variations, specifying that for this purpose the introduction of new parameters able to account for modifications of the attractor states would be required (page 29 of the revised manuscript).

C5. The authors added fig S2 which demonstrates an autocorrelation in the “noise” signal, but I do not understand why this is evidence for coupling as is claimed. There are many types of correlated noisy signal which exhibit temporal correlations: it is essential to clearly demonstrate the claimed coupling. Incidentally the meaning of lag should be clarified so that the reader can understand how long are the correlations (in seconds).

R5. We reported in Supplementary Figure S2 the analysis showing autocorrelation in the conductance time trace signal. We added this information to show that the conductance time trace cannot be modeled simply by superimposing uncorrelated noise to a deterministic trajectory. Indeed, in a stationary state no autocorrelation is expected if deterministic behavior and noise/jumps are uncoupled. Even if correlation can be induced by adding a noise signal which exhibits temporal correlation, the autocorrelation observed in experimental data spontaneously arises by describing the system in terms of an OU process. The description in terms of an OU process with intertwined deterministic and stochastic dynamics is further corroborated by the analysis of the trend of the autocorrelation function of the output signal (in a stationary state) as a function of lags, reported in Figure R4. Notably, experimental results are in accordance with an exponential behavior of the autocorrelation function as a function of lags, as expected from an OU process⁷. Note that fitting of the autocorrelation function was performed by considering correlations that are significant with a 95 % confidence interval. Here, the lag relies on the sampling rate of the measurements and corresponds to $\Delta t \sim 0.63$ s.

Figure R4. | Autocorrelation plot of experimental data. a. Example of an experimental conductance time trace of a network in a stationary state sustained by an applied bias voltage of 3.6 V and b. corresponding autocorrelation plot. The autocorrelation plot, where it is possible to see that autocorrelation decreases as a function of lags, shows that the conductance time trace cannot be simply modeled by superimposing a (uncorrelated) noise to a deterministic trajectory. Indeed, in a stationary state no autocorrelation is expected if deterministic and stochastic dynamics are not coupled (i.e., simply adding noise to a deterministic trajectory in a stationary state does not result in autocorrelation). Note that the exponential trend of the autocorrelation function of the signal as a function of lags (in a stationary state) agrees with the expected autocorrelation (exponential) from an OU process.⁷ The fitting of the autocorrelation function (red line) was performed by considering correlations that are significant with a 95 % confidence interval (left side of the vertical dashed line). The lag relies on the sampling rate of the measurements and corresponds to $\Delta t \sim 0.63$ s.

Changes in the manuscript:

- The above-mentioned discussion was added in the revised manuscript, on page 6 and 7. The related Supplementary Figure S2 and the caption text have been revised and updated with here reported Figure R4, showing evidence of the exponential dependence of the autocorrelation function on lags, further supporting our modeling approach based on the OU process.
- The meaning of lag has been clarified in the caption of the revised Supplementary Figure S2, so that the reader can now understand how long the correlations are. We have also included information about the sampling rate, directly related to the lag temporal length, in the Methods section (page 32)

C6. C12: The authors state in R12 that their aim is to develop “a *novel modeling framework* able to describe, in one compact equation ... implementation of unconventional computing paradigms”, but then state “Due to the parabolic nature of the potential, we cannot expect bifurcations or transition between stable states”, which brings in to question where emergent behaviour is present even in the model.

R6. According to the concept of “emerging behavior” of self-organizing memristive networks discussed in response R2, our scope is to show that it is possible to describe the emergent behavior of neuromorphic nanowire networks with a single equation model. Bifurcations or transitions among stable states are not expected due to the parabolic nature of the potential (these effects are not inherent to a linear mean-reverting type of OU process).

Changes in the manuscript: Besides clarifying the concept of “emerging behavior” in these systems, we clarified in the revised manuscript that due to the parabolic nature of potential our modeling approach does not endow bifurcations and transitions among coexisting deterministic stable states (page 18). In addition, as detailed in response R4, we clarified in the revised manuscript that the proposed model can be further expanded by introducing transitions between coexisting states by introducing new parameters able to account for modifications of the attractor states (page 29 of the revised manuscript).

C7. C13: Here the authors again discuss their model in their rebuttal and not experiments. My original question remains answered: “Can the authors identify features of the *experimental* data which could not be explained by a simple model of a voltage controlled resistor (with sigmoidal dependence on V) together with independent noise / jumps?”. To me this is critical to the paper and is the essence of my original questions i.e. what evidence is there in the experiments to support the authors’ claims?

I was really confused by the comments in the rebuttal about “adding constant noise” in the model – surely the noise ought to be coupled to g ?

R7. We have further strengthened the connection between the experimental system and the proposed modeling approach as discussed in R5 and R11, further supporting the description of the system using an OU process.

Concerning the comment about “adding constant noise,” our purpose was to explain that we considered an OU process where the noise amplitude σ does not depend on g . Note that this does not mean that stochastic and deterministic dynamics are not intertwined, since fluctuation in the conductance time trace still depends on the intertwining of noise and jumps with the (deterministic) mean reverting process (as described in eq. 1 of the manuscript). This is the spirit of an Ito-type stochastic differential equation. As the reviewer cleverly points out, we are indeed modeling our system as a voltage controlled resistor (see Eq.(2)). However, what we are also providing is an explanation about its voltage dependence, which is the result of a balance process, and the combined action of stochastic effects (Eq. (1)). As suggested also by the reviewer, it is not clear how a simple sigmoidal resistance can deal with the mean-reverting property.

Changes in the manuscript: Besides strengthening the connection between experimental system and the proposed modeling approach (according to changes in the manuscript discussed in R5 and R11), we clarified in the revised manuscript that the assumption of σ independent from g does not mean that stochastic and deterministic dynamics are independently described (page 7).

C8. C14: The authors response and the new figure focus on results from the simulations. My questions were about the experimental results.

R8. In the previous response, it was discussed that not only the attractor state but also network dynamics are affected by the applied voltages, showing the reversion speed θ of the OU process as a function of the bias voltage as Supplementary Figure S8 (now Supplementary Figure S9). Notably, these results come from modeling but are the results of the fitting of experimental data reported in Figure 2b.

Changes in the manuscript: It has been clarified (in the caption of Supplementary Figure S8, now Supplementary Figure S9) that the θ dependence on the bias voltage has been obtained by interpolating experimental data reported in Figure 2b with the proposed modeling approach.

C9. C15: The new figure S1 shows that the conductance of the system changes to a new value when a different voltage is applied. I do not understand how this data demonstrates the presence of an attractor i.e. stable states for *a wide variety of starting conditions of the system* (as per the Wikipedia definition). Maybe there is an alternative definition of attractor, but if so it should be clear to the reader.

R9. The new Supplementary Figure S1 is an example of how the system tends to the same stable state (within certain limits of course) when stimulated with the same voltage bias, independently of the starting condition of the system. This is in line with the definition of attractor state. More in detail, in our experiments reported in Supplementary Figure S1 we showed that a 3V stimulation drives the

system to the same attractor state (conductance level) both in case the system is starting from an initial condition that is the attractor state sustained by a 1.5 V bias (panel a) and in case the system is starting from an initial condition that is the attractor state at 0 V (panel b). In other words, under the same voltage bias stimulation the system tends to the conductance levels (state “3.” Panel a, and state “5.” in panel b), even if the system is starting from different conductance states (state “2.” in panel a and state “4.” in panel b). Even if this represents only an example, further evidence of the presence of voltage dependence attractor states can be found in our previous work (ref.⁸), where reservoir computing was experimentally implemented by exploiting the so-called “fading memory” property effects of NW networks. Indeed, the “fading memory” relies on the fact that the state of the system depends only on recent-past inputs (i.e., the conductance state depends on the history of stimulation in the short-term), but it is independent of distant-past inputs (i.e., the system tends to converge to the same state – attractor state – in the long-term, independently from initial conditions).

Changes in the manuscript: We have further clarified the above-mentioned aspects in the revised manuscript (page 6).

C10. C16: If I understand the response correctly, the authors are agreeing that they treat the noise and deterministic trajectory separately. The manuscript needs to include a demonstration of the intertwining that is claimed to exist between these terms.

R10. We agree with the reviewer that the sentence we wrote in the manuscript: “*The stochastic component of the neuromorphic network can be disentangled from the deterministic dynamics by analyzing the noise and jumps in the conductance when the system operates in the stationary state*” was somehow misleading. Indeed, in our work we analyzed noise and jumps in the stationary state as it is usually done in the case of stochastic dynamic systems (to avoid initial transient dynamics), but this does not mean that we treat noise and deterministic trajectories separately (in this sense, the term “disentangled” in the previously mentioned sentence was misleading). Please also refer to responses R2, R5, R7 and R11 for a detailed discussion on the intertwined action of deterministic and stochastic dynamics.

Changes in the manuscript: According to the above discussion, we have revised the corresponding sentence on page 12. Furthermore, see related changes in the manuscript related to responses R5, R7 and R11. In the revised manuscript we more correctly used the term “disentangling” to describe only the separation of jumps from noise (page 12), according to literature (refer for example to ref.⁹).

C11. C17: the plots provided are much clearer, and I see the experimental and simulated data in Fig S4 are similar. However I do not understand why the data in Fig S7 appear different to those in Fig S4 or why equivalent experimental data are not presented alongside Fig S7 – experimental data showing reversion to the mean after a jump would be invaluable in supporting the author’s claims.

R11. In Supplementary Figures S4, we show the dG/dt signal while in Supplementary Figure S7 (now Supplementary Figure S8) we reported the time trace of the evolution of the internal memory state of the system g (Supplementary Figure S7, now S8, is an enlarged view of Figure 6c of the manuscript). In the first case, the derivative of the conductance is shown, while in the second case the internal memory state g that is proportional to the conductance (according to eq. (2)) is reported. Thus, the two figures appear to be different since two different quantities are reported there. To further clarify this aspect, we provide in Figure R5 an additional direct comparison between the experimental (panel a) and simulated (panel c) conductance time traces. The observed qualitative agreement between experimental data and simulations further supports our modeling approach. As requested by

the reviewer, we also provide an example of experimental data showing the effect of the reversion to the mean after a jump in the conductance trace in Figure R5c. This experimentally observed behavior can be described by our model, as reported in Figure R5d showing reversion to the mean after a jump in a simulated conductance time trace. Together with the new experimental analysis reported in response to comment C5, these observations further support the intertwining between deterministic and stochastic effects in experimental data.

Figure R5. Direct comparison of experimental data and modeling. **a.** Experimental conductance time trace and **b.** experimental example showing the reversion to the mean effect after a jump in the conductance time trace. **c.** Modeled conductance time trace and **d.** modeled example showing the reversion to the mean effect after a jump in the conductance time trace. Experimental and modeled time traces for a stationary state sustained by an applied constant voltage of 3.6 V are reported.

Changes to the manuscript:

- We clarified in the revised manuscript that the intertwined action of stochastic and deterministic effects endowed in our modeling approach results also in qualitative agreement when considering the direct comparison of experimental and modeled conductance time traces in the stationary state (page 15).
- A new supplementary Figure reporting a direct comparison of experimental and modeling results, including also an experimental and modeling example showing the reversion to the mean effect after a jump in the conductance time trace, (here reported as Figure R2) has been added in the revised manuscript as new Supplementary Figure S5. A sentence recalling this new supporting figure has been added at page 15 of the manuscript.

C12. C18: The authors have clarified to some extent which data is experimental and which simulated but, especially for the new plots, this is still unclear in places e.g. Figs 8 and 9, S1 and S2. Please make sure this is clear at the beginning of every caption.

R12. We thank the reviewer for pointing out this aspect, we clarified which data is experimental and which is simulated in all figure captions.

Changes in the manuscript: We have clearly pointed out in Figures 8, 9, Supplementary Figures S1, S2, S3, S16, S17 whether reported results are experimental or simulations.

C13. I had to re-read the reservoir computing section multiple times before I was *nearly* sure that the computation had been simulated – the text needs to make this much clearer, and there should be a clear statement about why similar computations have not been carried out experimentally.

R13. We clarified through the text that computation had been simulated, specifying that further experimental activities out of the scope of this work are required to experimentally investigate computing capabilities of these systems. This is just a proof of concept that helps to understand the possible impact of our work, as requested by reviewers during previous iterations.

Changes in the manuscript:

- We have clarified at page 23 of the revised manuscript that a multiplexed reservoir computing scheme has been implemented through simulations. Also, it was clarified in related figure captions that results are from simulations, according to response R13.
- We have also clarified in the revised discussion paragraph (page 29) that, even if the here reported model with a limited number of assumptions can be adopted to explore *in silico* different implementations of the reservoir computing paradigm through simulations, further experimental activities (out of the scope of this work) are required to experimentally investigate computing capabilities of these systems.

References

1. Milano, G., Cultrera, A., Boarino, L., Callegaro, L. & Ricciardi, C. Tomography of memory engrams in self-organizing nanowire connectomes. *Nat Commun* **14**, 5723 (2023).
2. Vahl, A., Milano, G., Kuncic, Z., Brown, S. A. & Milani, P. Brain-inspired computing with self-assembled networks of nano-objects. *J Phys D Appl Phys* **57**, 503001 (2024).
3. Diaz-Alvarez, A. *et al.* Emergent dynamics of neuromorphic nanowire networks. *Sci Rep* **9**, 14920 (2019).
4. Pilati, D., Michieletti, F., Cultrera, A., Ricciardi, C. & Milano, G. Emerging Spatiotemporal Dynamics in Multiterminal Neuromorphic Nanowire Networks Through Conductance Matrices and Voltage Maps. *Adv Electron Mater* (2024) doi:10.1002/aelm.202400750.
5. Montano, K., Milano, G. & Ricciardi, C. Grid-graph modeling of emergent neuromorphic dynamics and heterosynaptic plasticity in memristive nanonetworks. *Neuromorphic Computing and Engineering* 0–22 (2022) doi:10.1088/2634-4386/ac4d86.
6. Milano, G. *et al.* Brain-Inspired Structural Plasticity through Reweighting and Rewiring in Multi-Terminal Self-Organizing Memristive Nanowire Networks. *Advanced Intelligent Systems* **2**, 2000096 (2020).
7. Gardiner, C. W. *Handbook of Stochastic Methods - For Physics, Chem, Nat. Sciences.* (Berlin, 1986).
8. Milano, G. *et al.* In-materia reservoir computing with a fully memristive architecture based on self-organizing nanowire networks. *Accepted Manuscript*.
9. Anvari, M., Tabar, M. R. R., Peinke, J. & Lehnertz, K. Disentangling the stochastic behavior of complex time series. *Sci Rep* **6**, 35435 (2016).

Point-by-point response

We thank the reviewers for their further comments that gave us the opportunity to further improve the overall quality of our manuscript. Given the importance we give to this work and based on the overall positive assessment of reviewers #1 and #2, we have refocused the entire manuscript by following the route toward publication suggested by reviewer #3. We believe that the current revised manuscript, proposing for the first time a modeling framework able to describe the behavior of neuromorphic nanowire networks with a minimal number of assumptions, now complies with the high publication standards required by *Nature Communications*.

Reviewer #2

C1. While the random nanowire network has its own flaws for reservoir computing, it is worthy investigating the so-called self-organizing capability. Other than that, I do not have any further comments.

R1. Despite agreeing with the reviewer that further investigation of self-organizing capabilities of these systems is worthwhile, the aim of our work is to provide for the first time a compact modeling framework able to describe the main features of deterministic and stochastic behavior of nanowire networks through a minimal number of assumptions. To avoid any issues, we have refocused the whole manuscript to clarify that the aim of our work is to show that the electrical response of neuromorphic nanowire networks can be modeled as a stochastic dynamic system endowing deterministic and stochastic effects. We have also included a new section in the discussion paragraph (page 28) describing experiments that would allow emergent behavior to be observed (see details in response R6 to reviewer #3).

Reviewer #3

C2. The authors have written a long response and made some further revisions to the manuscript, but they have not addressed the concerns I have raised. The authors repeatedly state that the emergent behavior and related effects of interest are evidenced by the results of their model. The model (eq 1) clearly includes the coupling between the effects of interest. But, as I emphasized in my previous reports, there is no evidence in the experimental data for coupling of these effects, and that is still the case in the revised manuscript.

The fact that a model that includes coupling can be used to generate effects similar to those in the experiments does not mean that the coupling is present in the experiments. A far simpler and more obvious explanation of the observed effects is that they occur through independent processes e.g. dynamical effects are observed in the conductance versus time and autocorrelation plots but this is not evidence for intertwined deterministic and stochastic dynamics as is claimed.

R2. We now better understand the reviewer's viewpoint, and we agree that our claims were potentially somehow misleading in previous versions of the manuscript. Rather than demonstrating that our nanowire networks are stochastic dynamical systems with intertwined deterministic and stochastic dynamics, the aim of our work is to provide for the first time a modeling approach able to describe the main features of the neuromorphic nanowire network behavior. In this terms, we propose a modeling approach based on a Ornstein-Uhlenbeck process with coupled deterministic and stochastic dynamics supported by *i*) the experimental observation of the reversion to the average trajectory after stochastic jumps in the experimental conductance time trace (examples in Supplementary Information

S4) and *ii*) the exponential decay of the autocorrelation function with the number of lags in the stationary state as expected for an OU process (Supplementary Information S5). Despite this compact model well describes main features of the experimental system, further experimental work is required to effectively demonstrate if coupling between deterministic and stochastic dynamics effectively occurs in the experimental system. In this context, the claim of the manuscript was carefully revised (refer to response to comment C6 of the reviewer).

Comments on some of the changes to the manuscript:

C3. every system that has interactions between components is not emergent

R3. We agree that not every system that has interactions between components endows an emergent behavior. Despite previous works discussed the evolution of the two-terminal conductance of nanowire networks in terms of emergent dynamics of the system (see for example ref. ¹), we avoid referring to the dynamics of the system as “emergent dynamics” in the manuscript in order to avoid any possible misunderstanding for the reader.

Changes in the manuscript: we avoid the use of “emergent dynamics” through the whole text.

C4. the demonstration in figure S2 that the conductance of the devices reach the same value each time 3V is applied does not convince me that there is an attractor - similar behavior could results from quite simple circuits and in this system would arise simply from sequential formation of a chain filaments.

R4. Even if modeling the long-run state of neuromorphic nanowire networks as a stable trajectory is supported by experimental observations showing that the system’s conductance tends toward the same steady state irrespective of the initial condition (as shown in Supplementary Figure S1), further experiments involving a wider variety of starting conditions of the system are required to experimentally confirm the presence of attractor states in the physical system. In any case, the proposed modeling approach can describe not only the deterministic evolution of the system towards steady states (refer to Figure 2a and c) but also the experimentally observed evolution of the steady state as a function of the applied voltage (Figure 2b).

Change in the manuscript: we clarified the above-mentioned aspects through the revised manuscript. To avoid any misunderstanding, we use the term “steady state” instead of “attractor state” through the whole manuscript, and we specified that further experiments involving a wider variety of starting conditions of the system are required to prove the presence of attractor states in the physical system (page 10).

C5. The new Fig S5(a) is the only evidence I see for mean reversion in experiments, but even there the jumps appear atypical (no data like that the right panel is visible in the left panel) and the mean values in the two panels are different.

R5. Noteworthy, the mean reversion effect in the manuscript can be observed not only in Supplementary Figure S5 but also in Figure 2 and Figure 5 where it is possible to observe that the experimental system mean-reverts towards the steady state condition under voltage stimulation. Supplementary Figure S5 (now Supplementary Figure S4 in the revised manuscript) shows qualitative agreement between the experimental conductance time trace and the time trace generated through modeling. While panel b and d are only a few examples suggesting mean-reversion after jumps, direct visualization of the mean-reverting property in the conductance time trace in both experimental data

and modeling (where mean reversion is imposed by the equation) is not straightforward due to the unavoidable co-occurrence of noise and jumps. The small variation of the mean values in panel a and b (quantifiable in the order of 0.005 mS) is ascribable to experimental variability (note that such small variation of the mean value is observed after more than 10000 s of measurements). In any case, the model endowing mean-reverting property allows the generation of a conductance time trace that, besides can statistically emulate stochastic effects observed in the experimental curve (refer to Figure 3 and 4, Supplementary Figure S3, Supplementary Figure S6), is also qualitatively very similar to the experimental time trace.

Changes in the manuscript: we better specified through the text (page 10 and 17) that the network behavior in experimental data reported in Figure 3 and 5 can be described by means of the mean reverting property of eq. (1). Also, we specified in Supplementary Figure S4 that the direct visualization of the mean-reverting property even in the modeled conductance time trace (where mean reversion is imposed) is not straightforward due to the unavoidable co-occurrence of noise and jumps (refer to panel c of Supplementary Figure S4).

C6. It is clear now that the reservoir computing aspect of the paper is done through simulations but the fact that simulations have to be used amplifies the concerns I have about the lack of coupling / complex dynamics in the experimental data.

R6. According to previous reviewer requests, we have extended our previous revisions our work by including simulations of computing with the aim of establishing a connection between the proposed model and the reservoir computing paradigm, quantitatively connecting deterministic and stochastic dynamics of the model with computing capabilities. As specified through the text (page 30), further work is required to experimentally investigate computing capabilities of these systems. However, our simulations show that the proposed model able to describe main features of the experimental system can be exploited to explore *in silico* different implementations of the reservoir computing paradigm, guiding further experiments.

C6. I can however see that the claimed coupling potentially might eventually be observable in nanowire networks and so I want to suggest a route toward publication. I wonder if the authors would refocus their paper on results from simulations and include a section that describes adaptations of the experiments that would allow emergent behavior to be observed. This would potentially avoid the issues in relation to the experimental data that I believe are misleading at present.

R6. We have refocused the entire manuscript according to the reviewer's suggestion, adding also a new section and new supplementary note describing experiments that would allow emergent behavior to be experimentally observed (as discussed in the new section and supplementary note, this requires advancements in the state-of-the-art characterization techniques). We carefully revised the claim of the manuscript to avoid any misunderstanding, clarifying that we don't aim to experimentally demonstrate that nanowire-based neuromorphic networks are stochastic dynamical systems but rather to show that neuromorphic nanowire networks can be modeled as stochastic dynamic systems with a minimal number of assumptions. Considering the reviewer viewpoint, we understand that this could be somehow misleading in the previous version of our work.

Changes in the manuscript:

- We have carefully revised the claim of our work in the abstract, introduction, and conclusion, and through the whole manuscript, focusing on modeling.
- We changed the title from "Self-organizing neuromorphic nanowire networks are stochastic dynamical systems" to "Self-organizing neuromorphic nanowire networks as stochastic

dynamical systems” to avoid any misunderstanding on the claim of our work already from the title.

- We better specified in section “*Self-organizing memristive networks as stochastic dynamical systems*” (page 6) that the aim of this work is to provide a unified framework based on an OU process with jumps that can be exploited for modeling deterministic and stochastic dynamics of neuromorphic nanowire networks (we agree with the reviewer viewpoint that the introduction of this section was somehow misleading in the previous version of the manuscript).

- We included the following new section in the discussion (page 28) describing experiments that would allow emergent behavior to be observed:

It is worth mentioning that the ultimate origin of the modeled network behavior is experimentally difficult to access due to the huge number of junctions involved. Indeed, understanding the origin of deterministic dynamics, noise and jumps of the conductance time trace necessarily implies high-resolution visualization of the spatially distributed electrical activity across the whole network. Even if it has been shown that it is possible to experimentally investigate the spatial conductance distribution over the network through electrical resistance tomography,^{2,3} this technique does not allow the spatial resolution required to unveil how switching events in single NW junctions impact the resulting network behavior. Other techniques such as the voltage contrast SEM imaging⁴, besides being applicable only to networks with limited size, require making measurements in vacuum, altering electrochemical effects governing resistive switching events in NW junctions and, thus, altering the resulting behavior and electrical response of the system. In any case, experiments in multiterminal configurations involving synchronous recording of activity in different network areas (such as the recording of voltage maps reported in ref. ⁵) and the experimental investigation on how power is spatially dissipated through the network (such as through lock-in thermography⁶) can provide further insights on how local activity impacts the resulting network behavior as well as on the coupling between deterministic and stochastic dynamics of the system. For this purpose, advancements in these techniques are required for improving their spatial and temporal resolution.

- We included the following detailed discussion on experiments that can be exploited to observe the network behavior as a new supplementary note (Supplementary Note 1):

The spatially distributed network activity can be accessed through characterization techniques that allows to measure local electrical properties of the networks and their evolution over time. For this purpose, techniques such as Electrical Resistance Tomography (ERT), voltage-contrast SEM imaging, conductive AFM, lock-in thermography, and multiterminal measurements can be exploited, as detailed in the following.

Electrical Resistance Tomography (ERT).^{3,7} This quantitative and non-scanning technique allows reconstruction of the spatial distribution of conductivity across the network from boundary electrical measurements. Despite this technique enables to map the conductivity of networks over large areas ($\approx 1 \times 1 \text{ cm}^2$), it has a very low spatial resolution ($\approx 2 \text{ mm}$) that does not allow to investigate how local effects in NW junctions reflects in the resulting network behavior. Furthermore, the long acquisition time required for mapping ($\approx 40 \text{ s}$) does not allow to investigate the origin of conductance fluctuations over time in the network with high temporal resolution.

Voltage-contrast Scanning Electron Microscopy (SEM) imaging.^{4,8} This scanning technique enables to acquire passive voltage contrast images that allow the observation of current

pathways through the network. Even if this technique endows high spatial resolution (estimated as ≈ 10 nm), the scanning area is limited (up to $\approx 100 \times 100 \mu\text{m}^2$). Since the resolution decreases while increasing the scanning area, this method does not allow to investigate the behavior of networks with more than a few NWs with single junction resolution. Also, it should be highlighted that SEM imaging usually requires vacuum conditions that can alter the resistive switching mechanism. Despite advancements in low-vacuum SEM systems, voltage-contrast imaging to observe current pathways in self-organizing networks of nano objects still have to be demonstrated in air at ambient pressure.

Conductive Atomic Force Microscopy (C-AFM).⁸ This scanning technique enables to measure local current flowing between the C-AFM tip and a reference electrode, allowing to evaluate the conductivity of specific network areas. Despite the high resolution of this technique (\approx few nm), the scanning area is limited ($\approx 50 \times 50 \mu\text{m}^2$). Furthermore, the long acquisition time of this scanning technique (\approx min) does not provide the temporal resolution for investigating physicochemical phenomena underlying conductance fluctuations observed in NW networks.

Lock-in thermography.⁹ This technique enables to evaluate the local infrared emission (that can be converted to temperature) of the network when electrically stimulated, allowing to observe current pathways through the network with a spatial resolution of $\approx 3 \mu\text{m}^2$ over a scanning area of $\approx 1 \times 1 \text{mm}^2$. However, the state-of-the-art of this technique does not allow to investigate the dynamic behavior of the system due to the long acquisition time (≈ 50 s). Also, it should be highlighted that in this case the measurand is represented by the photons emitted by the network that are then used to infer local electrical properties, enabling to visualize only main conductive pathways that dissipate more power. In any case, advancements in the resolution of this technique can provide information on how power is dissipated through the network, an aspect that is directly related to the impact of network areas on the resulting network behavior.

Multiterminal measurements.¹⁰⁻¹⁵ Multiterminal measurements have been widely exploited in NW networks mainly to nonlinearly map an input pattern in an output signal in the context of reservoir computing.¹⁰⁻¹⁴ In all these contexts, the network was treated as black-box for processing an input signal for computing purposes. Recently, it has been shown that a multiterminal approach enabling synchronous probing of local electrical activity (16 electrodes over a sample of $\approx 1 \times 1 \text{cm}^2$) can be exploited to extract information on the local network activity and on its impact on the resulting network behavior.¹⁵ By synchronously measuring the voltage over floating network distributed across the network and by reconstructing voltage maps and conductance matrices, it has been shown that multiterminal measurements can be exploited to monitor the spatial distribution of nonlinear activity, thus enabling to observe the impact of network areas on the resulting network behavior. Despite further advancements in multiterminal characterization are required to enhance the spatial resolution of this technique (for this purpose, multiterminal setups with an increased number of probing electrodes are required), this represents a promising technique for investigating how the resulting network behavior arises from local network activity under electrical stimulation.

References

1. Diaz-Alvarez, A. *et al.* Emergent dynamics of neuromorphic nanowire networks. *Sci Rep* **9**, 14920 (2019).
2. Milano, G., Cultrera, A., Boarino, L., Callegaro, L. & Ricciardi, C. Tomography of memory engrams in self-organizing nanowire connectomes. *Nat Commun* **14**, 5723 (2023).
3. Milano, G. *et al.* Mapping Time-Dependent Conductivity of Metallic Nanowire Networks by Electrical Resistance Tomography toward Transparent Conductive Materials. *ACS Appl Nano Mater* acsanm.0c02204 (2020) doi:10.1021/acsanm.0c02204.
4. Manning, H. G. *et al.* Emergence of winner-takes-all connectivity paths in random nanowire networks. *Nat Commun* **9**, 3219 (2018).
5. Pilati, D., Michieletti, F., Cultrera, A., Ricciardi, C. & Milano, G. Emerging Spatiotemporal Dynamics in Multiterminal Neuromorphic Nanowire Networks Through Conductance Matrices and Voltage Maps. *Adv Electron Mater* **10**, (2024).
6. Li, Q. *et al.* Dynamic Electrical Pathway Tuning in Neuromorphic Nanowire Networks. *Adv Funct Mater* **30**, 2003679 (2020).
7. Milano, G., Cultrera, A., Boarino, L., Callegaro, L. & Ricciardi, C. Tomography of memory engrams in self-organizing nanowire connectomes. *Nat Commun* **14**, 5723 (2023).
8. Nirmalraj, P. N. *et al.* Manipulating Connectivity and Electrical Conductivity in Metallic Nanowire Networks. *Nano Lett* **12**, 5966–5971 (2012).
9. Li, Q. *et al.* Dynamic Electrical Pathway Tuning in Neuromorphic Nanowire Networks. *Adv Funct Mater* **2003679**, 2003679 (2020).
10. Sillin, H. O. *et al.* A theoretical and experimental study of neuromorphic atomic switch networks for reservoir computing. *Nanotechnology* **24**, (2013).
11. Avizienis, A. V. *et al.* Neuromorphic atomic switch networks. *PLoS One* **7**, (2012).
12. Demis, E. C. *et al.* Atomic switch networks—nanoarchitectonic design of a complex system for natural computing. *Nanotechnology* **26**, 204003 (2015).
13. Lilak, S. *et al.* Spoken Digit Classification by In-Materio Reservoir Computing With Neuromorphic Atomic Switch Networks. *Frontiers in Nanotechnology* **3**, 1–11 (2021).
14. Diaz-Alvarez, A., Higuchi, R., Li, Q., Shingaya, Y. & Nakayama, T. Associative routing through neuromorphic nanowire networks. *AIP Adv* **10**, (2020).
15. Pilati, D., Michieletti, F., Cultrera, A., Ricciardi, C. & Milano, G. Emerging Spatiotemporal Dynamics in Multiterminal Neuromorphic Nanowire Networks Through Conductance Matrices and Voltage Maps. *Adv Electron Mater* (2024) doi:10.1002/aelm.202400750.

Point-by-point response

We thank the reviewers once again for their exceptional efforts in improving the quality of the submitted work. Your comments and suggestions have contributed enormously to the final version of the manuscript. We also thank the editor for giving us this new opportunity to shed further light on some terms and concepts invoked along our work.

Reviewer #2

In the revised manuscript, the authors made improvement, mainly on the previously misleading terms and concepts, to address reviewer 3's comments. I agree with those changes and suggest this latest version to be published.

We deeply thank the reviewer for his/her positive assessment of our revised work.

Reviewer #3

I think the revisions made are a significant improvement and in particular that the emphasis on modelling much more accurately reflects the data presented. If a few more relatively small things were clarified I think the manuscript could be accepted.

We thank the reviewer for his/her comments concerning our revised version of the submitted work. They have greatly helped us to improve the quality of the manuscript.

C1. Title: the authors have changed just one word in the title, but does not convey the change in content of the manuscript. I think a change in title is needed, and in particular the role of modelling in the paper probably needs to be highlighted.

R1. Concerning the reviewer's suggestion, the main point is that if we put so much emphasis on the "modelling" subject we think we are missing a large part of the experimental work carried out. Experiments are also essential sections of the paper's content, since the proposed modelling is based on experimental results. In any case, we highlighted the role of modelling in the abstract section. We have modified the corresponding paragraph in the abstract with: "... *neuromorphic nanowire network behavior can be modelled as an Ornstein-Uhlenbeck process ...* "

C2. Line 33: the sentence in the abstract beginning "Showing that information processing capabilities can...":the word can suggests much more certainty than is warranted given that it was not possible to show these effects experimentally.

R2. Following the reviewer's suggestion, the above-mentioned sentence was removed, and the abstract was shortened according also to the editor's requests.

C3. Line 80: The phrasing of "The model was exploited to investigate the influence of deterministic and stochastic dynamics on the information processing capabilities of the system..." makes it seem that stochastic dynamics might be a good thing for computation, but it is shown later on that the stochasticity reduces computational performance, so this is misleading. In this context the following sentence about embracing stochasticity then doesn't seem to make sense. Please clarify these statements. Similar clarifications of the role of stochasticity are needed in line 670 and in the remainder of that paragraph, which is unchanged from the previous version of the manuscript. If this discussion remains in the paper it needs to be clearer that the stochastic dynamics of the model and intertwining effects are not observed in the experiments. The simpler alternative would be to remove

this discussion. The same issue arises again in line 689 and line 692 where “connect ... stochastic dynamics with computing capabilities” and “embracing stochasticity” are misleading.

R3. We have revised the text as detailed below:

Line 80: we changed the word “influence” with “impact”. The last part of the following sentence was revised, the sentence was shortened. Refer to page 4 of the revised manuscript.

Line 670: Even if at first sight the random nature of the output signal seems to be detrimental for computing when adopting a deterministic perspective, it was shown that stochastic dynamics can indeed be practically exploited adding a new dimension for the implementation of stochastic learning rules, as well as for the hardware realization of random number generators, physical unclonable functions and chaotic/stochastic computing systems by taking advantage of the material substrate as the underlying source of randomness. We specified in the cited paragraph that “...*the inherent capability of the physical system to approximate the dynamic evolution of the stochastic differential equations ...*”, to clarify that the system evolution approximate the OU process. Refer to page 31 of the revised manuscript.

Line 698: we have revised the sentence, clarifying that “... *the compact model description can be exploited to quantitatively understand the impact of deterministic and stochastic dynamics on computing capabilities ...*”. Refer to page 32 of the revised manuscript.

Line 692: we have shortened the sentence, avoiding the use of “embracing stochasticity” in the sentence. Refer to page 32 of the revised manuscript.

C4. Line 139: “... jumps have the same physical origin...” this is surely speculation. Please provide evidence or make it clear this is speculation.

R4. We have revised the text, clarifying that “*In this scenario, jumps are expected to have the same physical origin...*” (page 6 of the revised manuscript)

C5. Line 226: the sentence “Even if modelling...” is too long and hard to understand

R5. This sentence has been shortened (page 10 of the revised manuscript).

C6. Line 251: “noise and jumps are not independent from the deterministic dynamics”. Please provide evidence or make it clear this is speculation or that it is only true in some versions of the model.

R6. We have clarified that noise and jumps are not independent from the deterministic dynamics in the proposed model approach, avoiding any misunderstanding. We have clarified that “...*in the proposed modeling approach noise and jumps are not independent from deterministic dynamics even in the stationary state, ...*” (page 12 of the revised manuscript). The deterministic dynamics provides the reversion mechanism necessary for keeping the random outcome bounded and correlated. We specified through the manuscript that this represents an approximation of the actual behavior of the system (the simplest approximation) at page 16 of the revised manuscript where we have now specified that “*Even if the OU modeling approach can represent the simplest approximation of the actual behavior of the experimental system ...*”

C7. Line 330: there is no need to introduce the exponent gamma, since $\gamma = 1$ this is a simple linear equation.

R7. We removed any reference to gamma in the revised equation (page 15 of the revised manuscript).

C8. Line 346: as pointed out in a previous report many other processes could give rise to an exponential decay of the correlation function so I do not accept that this is evidence for intertwined deterministic and stochastic effects. The authors should remove phrasing that obscures the fact that experimental evidence for intertwining is very limited.

R8. We agree with the reviewer that other processes could give rise to an exponential decay of the correlation function. In this context, the OU modelling approach exploited in our work represents the simplest approximation of the actual behaviour of the experimental system including both deterministic and stochastic behaviour. We further clarify this concept through the manuscript, and at page 16 of the revised manuscript where we have now specified that “*Even if the OU modeling approach can represent the simplest approximation of the actual behavior of the experimental system ...*”. In this context, we have also revised Supplementary Figure S5 (panel b) since a mistake on the confidence interval was present (this does not affect any conclusion).

C9. Page 23: there is no comparison of the obtained computational performance with the literature. The authors should at least add a few sentences that compare with performance obtained from other self-organised networks. Ref 58 appears to discuss time-multiplexed reservoir computing so that seems especially relevant.

R9. We added a new paragraph in the revised manuscript (page 25) comparing computational performances of our system with other self-organizing systems, adding also new relevant references.

C10. Page 28: some acknowledgement / discussion of other papers that discuss similar imaging techniques is needed e.g. Gronenberg et al, Adv. Funct. Mater. 2024, 2312989.

R10. We have revised the discussion in the manuscript specifying that “...techniques such as voltage contrast and resistive contrast scanning electron microscopy imaging” can be exploited, adding also the reference suggested by the reviewer (page 28 of the revised manuscript). The reference list was updated by including also most recent and relevant works.

C11. Line 642: “further work is required to experimentally investigate computing capabilities of these systems.” appears to ignore the substantial amount of work done on computing in other self-organised systems. The authors should acknowledge those results at this point in the paper.

R11. According to the reviewer comment, we have included in the revised manuscript (page 30) a paragraph mentioning experimental implementations of reservoir computing in self-organizing systems, also adding proper relevant references.

C12. The authors have introduced some new typographical and wording errors that need to be fixed e.g.

Line 75: “as dynamical system”

Line 206 “as similarly performed in”: performed seems to be the wrong word choice

Line: 342 “Despite further experiments are required to prove” needs rewording

Line 407 “due to changes in the applied voltage results in...”

Line 431 “is usually transduced in”

Line 443: “of the networ.”

Line 662: one of “fixed-point” and “steady state” seems redundant

Line 636: “stimulations conditions”

Line 639: “on computing performances.”

R12. Typographical and wording errors have been fixed in the revised manuscript.

In this manuscript Milano et al describe the dynamical properties of nanowire networks within a framework of emergent behaviour and neuromorphic computing, and attempt a holistic description of deterministic and stochastic effects based on Ornstein-Uhlenbeck (OU) processes. This is an interesting topic and is very relevant to a growing literature on nanoscale networks. Unfortunately I have a number of concerns.

The aim is to “demonstrate that nanowire-based neuromorphic networks are stochastic dynamical systems where the signals flow relies on the *intertwined* action of deterministic and random factors” and there is an emphasis throughout the manuscript on the importance of the internal dynamics of the system and of emergent behaviour. My understanding is that the intention is to demonstrate the rich dynamics, which might result from the coupling (intertwining) of the three terms in eq 1. Hence my main concern is that I do not see clear evidence in the experimental data for internal dynamics of the system, emergent behaviour, or for coupling.

Perhaps even more fundamental to the manuscript is whether OU processes, which are typically walk type processes, can lead to the kind of dynamics and emergent behaviour that might be useful for neuromorphic computing. It seems to me essential to discuss this point in the manuscript. Descriptions of stochastic dynamical systems in the literature make it clear that OU systems can be interesting as noise can stabilize unstable equilibria, shift bifurcations, and can lead to transitions between coexisting deterministic stable states. But such effects are not clearly demonstrated in the manuscript at the moment.

Throughout the paper it appears that the deterministic effects are analysed quite separately from the noise and jumps. For sure there are transient effects – when the input is changed, the conductance slowly evolves towards a steady state value – but I do not see evidence for coupling of the conductance changes to the noise or jumps. In fact Fig 1 seems to make it clear that the noise / jumps are *independent* of the conductance (and in Fig 5 the noise / jumps appear to simply be superimposed onto the deterministic behaviour). If they are dependent this needs to be clearly shown. Can the authors identify features of the experimental data which could not be explained by a simple model of a voltage controlled resistor (with sigmoidal dependence on V) together with *independent* noise / jumps?

In what way does conductance g represent internal *memory* of the system? The conductance of the system is controlled by the applied voltage (Fig 2b showing sigmoidal behaviour). On p19 it is stated that “enhanced dynamics of the internal memory state (i.e., higher dynamical range of g) can be observed...”, which relates to the data in Fig 7. From the data presented it appears that the higher dynamical range results simply from the differences in the gradient of the sigmoidal response to the applied triangle wave i.e. the deterministic response has a steep gradient at 3.6V and almost zero gradient at 5V. If these oscillations were related to the internal dynamics of the network, would they not have similar timescales to the transients rather than the period of the driving waveform?

A related point is the use of the word “attractor”. From wikipedia: “In the mathematical field of dynamical systems, an attractor is a set of states toward which a system tends to evolve, *for a wide variety of starting conditions of the system.*” Rather than demonstrating attractor states, the presented experimental data seems to show that each applied input voltage (i.e. the starting conditions) tunes the system to a *different* state. It would be great if the authors could clearly show that the system exhibits attractor states.

Similar points can be made in relation to the mathematical argument presented. It is stated that “when jumps are included, the stochastic differential equation needs to be solved numerically through the Euler–Maruyama method (details in Methods).” This would indeed be the case if complex dynamics were observed, but to the best of my understanding in this paper the deterministic part of the equation is simply integrated to give the average rate of change of the conductance (θ) and this is done completely separately from the analysis of the noise (eq 7). Perhaps this point is intended to be clarified in the methods but I found the text on p26-27 very unclear. This point is vital for the paper so it is really important that this argument is clarified and a careful summary of the detailed argument was added to the main text. It would help a lot if figures were added illustrating the process described on p26 using

real experimental data. Similarly I found the text on p27 describing the construction of the energy U to be unclear with some specific points of confusion (how can a derivative with respect to conductance be equal to a derivative with respect to conductance (the dimensions are surely not correct) and where does this equation come from?).

I found it difficult to understand the nature of the noise and jumps as the data is not presented on scales that allow examination of the data. The only figure that shows the noise and jumps clearly is Fig 6c, and there the noise appears to be almost sinusoidal, which seems to be quite different to that in Figs 1 and 5a, although the scale in earlier figures makes it difficult to be sure. Even in Fig 6c it would help the reader a lot if there were zoomed in plots that allow the response to a jump to be visualized.

Throughout pages 17-19 I was unable to tell which data / comments related to experiments and which to simulations. This needs to be clarified in all the captions as well as the text. I am guessing, but I think perhaps Fig 6c may be from the simulations, in which case comparison with similar zoomed in plots of the experimental data are essential to understand how reliable the characterization of noise and jumps are, and to allow an understanding of whether the simulation and experiments are similar or not. I suggest that the authors show segments of data from Figs 3a, d and g on much shorter timescales. I was surprised to find a comment in the methods section that Fig 7 is from simulations rather than experiments and I wonder why comparable experimental data are not presented. A clear comparison between experimental and simulated data would help the reader enormously.

Some more technical points:

1. The data presented in fig 1d and e does not clearly show that an equilibrium is reached and so data should be shown for a longer time period.
2. How can the noise in Fig 1f be clearly distinguished from the jumps? Isn't it possible that the noise is just smaller jumps?
3. It is stated that Fig 2b shows a long tailed distribution and (in the methods) that logarithmic bins are used. Neither of these appears to be correct and furthermore the logarithmic y axis in b makes it impossible to compare with Fig 2e. Fig 2h does not look like an exponential – why is the data not shown on log linear scales?
4. The equation for Poisson processes on p14 surely cannot really be $N(t)=t$? What is the meaning of $\lambda=1$?
5. Data should be presented to allow the MLE fitting to be understood.

Finally, while the manuscript is generally clearly written, if the authors intend to revise their manuscript they should also revise some of the choices of wording. Examples include:

6. Is software really aim of the neuromorphic computing community?
7. Terms like “self-organized synergy” and “stochastic emerging behavior” seem meaningless and should be removed.
8. “Reversion speed” is not explained – it seems to be simply a rate of change.
9. Why is the network conductance “effective”?

The authors have made some significant revisions to the manuscript, including adding a new section on computation, but I was unable to see changes that address the majority of my original comments.

As with the original manuscript my main concerns relate to the central claim in the abstract “we demonstrate that *nanowire-based neuromorphic networks* are stochastic dynamical systems where the signals flow relies on the *intertwined* action of deterministic and random factors” and comments in the conclusions about the importance of emergent behavior. In my view, since the nanowire networks are experimental devices, to justify their claims the authors need to provide *experimental* data that support their claims, and specifically data that support the claims that the deterministic and random factors are intertwined, and lead to emergent behavior. Much of the manuscript relates instead the authors’ model and the connection to the experimental system is not nearly clear enough.

Below I clarify why I do not think my original comments have been addressed. Note that while my comments mostly refer to the rebuttal I want to emphasize that these points really *have* to be clear *in the manuscript*.

C11: In R11 the authors state “that emergent behavior and coupling are inherent to the *assumed description*”. I have no doubt that *the model* can exhibit emergent behavior, but I do not see evidence for this in the experiments. I think it is essential that the authors clarify what evidence there is for these effects *in the experimental data*.

I was confused by R12 (“the scope of the work was not to demonstrate shift bifurcations nor the transitions between coexisting states”) which seems to suggest that even the model is in a regime where emergent behavior is not observed. Is that correct? All of this needs to be clearly discussed in the manuscript.

The authors added fig S2 which demonstrates an autocorrelation in the “noise” signal, but I do not understand why this is evidence for coupling as is claimed. There are many types of correlated noisy signal which exhibit temporal correlations: it is essential to clearly demonstrate the claimed coupling. Incidentally the meaning of lag should be clarified so that the reader can understand how long are the correlations (in seconds).

C12: The authors state in R12 that their aim is to develop “a *novel modeling framework* able to describe, in one compact equation ... implementation of unconventional computing paradigms”, but then state “Due to the parabolic nature of the potential, we cannot expect bifurcations or transition between stable sates”, which brings in to question where emergent behaviour is present even in the model.

C13: Here the authors again discuss their model in their rebuttal and not experiments. My original question remains answered: “Can the authors identify features of the *experimental* data which could not be explained by a simple model of a voltage controlled resistor (with sigmoidal dependence on V) together with independent noise / jumps?”. To me this is critical to the paper and is the essence of my original questions i.e. what evidence is there in the experiments to support the authors’ claims?

I was really confused by the comments in the rebuttal about “adding constant noise” in the model – surely the noise ought to be coupled to g?

C14: The authors response and the new figure focus on results from the simulations. My questions were about the experimental results.

C15: The new figure S1 shows that the conductance of the system changes to a new value when a different voltage is applied. I do not understand how this data demonstrates the presence of an attractor i.e. stable states for *a wide variety of starting conditions of the system* (as per the Wikipedia definition). Maybe there is an alternative definition of attractor, but if so it should be clear to the reader.

C16: If I understand the response correctly, the authors are agreeing that they treat the noise and deterministic trajectory separately. The manuscript needs to include a demonstration of the intertwining that is claimed to exist between these terms.

C17: the plots provided are much clearer, and I see the experimental and simulated data in Fig S4 are similar. However I do not understand why the data in Fig S7 appear different to those in Fig S4 or why equivalent experimental data are not presented alongside Fig S7 – experimental data showing reversion to the mean after a jump would be invaluable in supporting the author's claims.

C18: The authors have clarified to some extent which data is experimental and which simulated but, especially for the new plots, this is still unclear in places e.g. Figs 8 and 9, S1 and S2. Please make sure this is clear at the beginning of every caption.

I had to re-read the reservoir computing section multiple times before I was *nearly* sure that the computation had been simulated – the text needs to make this much clearer, and there should be a clear statement about why similar computations have not been carried out experimentally.

I think the revisions made are a significant improvement and in particular that the emphasis on modelling much more accurately reflects the data presented. If a few more relatively small things were clarified I think the manuscript could be accepted.

Title: the authors have changed just one word in the title, but does not convey the change in content of the manuscript. I think a change in title is needed, and in particular the role of modelling in the paper probably needs to be highlighted.

Line 33: the sentence in the abstract beginning “Showing that information processing capabilities can...”: the word can suggests much more certainty than is warranted given that it was not possible to show these effects experimentally.

Line 80: The phrasing of “The model was exploited to investigate the influence of deterministic and stochastic dynamics on the information processing capabilities of the system...” makes it seem that stochastic dynamics might be a good thing for computation, but it is shown later on that the stochasticity reduces computational performance, so this is misleading. In this context the following sentence about embracing stochasticity then doesn’t seem to make sense. Please clarify these statements. Similar clarifications of the role of stochasticity are needed in line 670 and in the remainder of that paragraph, which is unchanged from the previous version of the manuscript. If this discussion remains in the paper it needs to be clearer that the stochastic dynamics of the model and intertwining effects are not observed in the experiments. The simpler alternative would be to remove this discussion. The same issue arises again in line 689 and line 692 where “connect ... stochastic dynamics with computing capabilities” and “embracing stochasticity” are misleading.

Line 139: “... jumps have the same physical origin...” this is surely speculation. Please provide evidence or make it clear this is speculation.

Line 226: the sentence “Even if modelling...” is too long and hard to understand

Line 251: “noise and jumps are not independent from the deterministic dynamics”. Please provide evidence or make it clear this is speculation or that it is only true in some versions of the model.

Line 330: there is no need to introduce the exponent gamma, since $\gamma = 1$ this is a simple linear equation.

Line 346: as pointed out in a previous report many other processes could give rise to an exponential decay of the correlation function so I do not accept that this is evidence for intertwined deterministic and stochastic effects. The authors should remove phrasing that obscures the fact that experimental evidence for intertwining is very limited.

Page 23: there is no comparison of the obtained computational performance with the literature. The authors should at least add a few sentences that compare with performance obtained from other self-organised networks. Ref 58 appears to discuss time-multiplexed reservoir computing so that seems especially relevant.

Page 28: some acknowledgement / discussion of other papers that discuss similar imaging techniques is needed e.g. Gronenberg et al, Adv. Funct. Mater. 2024, 2312989.

Line 642: “further work is required to experimentally investigate computing capabilities of these systems.” appears to ignore the substantial amount of work done on computing in other self-organised systems. The authors should acknowledge those results at this point in the paper.

The authors have introduced some new typographical and wording errors that need to be fixed e.g.

Line 75: “as dynamical system”

Line 206 “as similarly performed in”: performed seems to be the wrong word choice

Line: 342 “Despite further experiments are required to prove” needs rewording

Line 407 “due to changes in the applied voltage results in....”

Line 431 “is usually transduced in”

Line 443: “of the networ.”

Line 662: one of “fixed-point” and “steady state” seems redundant

Line 636: “stimulations conditions”

Line 639: “on computing performances.”